# Conditional Quantile Adjusted Conformal Prediction for Time Series

Cheng Yu [1]    Zhoufan Zhu [2 3 *]    Ke Zhu [4]

## Abstract

Conformal prediction is challenging for time series with time-varying conditional distributions. Existing sequential conformal methods can yield volatile, non-nested prediction intervals due to noisy tail conditional quantile estimation and quantile crossing issue. To overcome these challenges, we construct prediction intervals for time series via a novel method called Conditional Quantile Adjusted Conformal Prediction (CQACP), which stabilizes sequential conformal calibration by modeling the conditional quantile curve of nonconformity score. At each time step, CQACP evaluates a base conditional quantile learner on a grid of quantile levels, and fits a Cornish–Fisher approximation parameterized by conditional moments of nonconformity score with monotonicity constraints. Asymptotically, we prove the conditional validity of the prediction interval under serial dependence and show improved conditional quantile estimation accuracy. Experiments on multiple real-world datasets demonstrate that CQACP maintains accurate coverage and produces smooth, narrow, and nested prediction intervals across different significance levels and prediction models.

## 1. Introduction

Reliable uncertainty quantification is a core requirement in sequential forecasting tasks, especially the situations where decisions depend not only on a point forecast but also on a calibrated assessment of risk (Sun et al., 2021; Jin & Candès, 2023; Gui et al., 2024). Prediction intervals provide a natural

link between point forecasts and risk assessment by characterizing the range of plausible future outcomes around a point prediction at a given significance level. However, in time series settings, constructing prediction intervals is especially challenging: The conditional distribution of the next outcome can change quickly over time, exhibit strong serial dependence, and display (conditional) heteroskedasticity, skewness, and heavy tails (Tsay, 2005). In such regimes, prediction intervals based on fixed parametric assumptions are often fragile, while purely empirical calibration can be unstable in the tails.

Conformal prediction provides a general framework for constructing prediction intervals with rigorous coverage guarantees, while allowing the employed prediction model to be a black box. Under exchangeability, conformal prediction yields finite-sample marginal coverage with minimal assumptions; however, classical exchangeability is violated in time series due to serial dependence and conditional heteroskedasticity. To overcome this challenge, a growing literature has developed conformal methods for sequential and dependent data, typically by updating the calibration threshold online or explicitly modeling the evolution of nonconformity scores. A representative and influential approach is Sequential Predictive Conformal Inference (SPCI; Xu & Xie, 2023). SPCI replaces empirical residual quantiles with conditional quantiles of the next nonconformity score given recent residual history, improving adaptivity under serial dependence and yielding asymptotic conditional validity under suitable regularity conditions. Note that the conformal prediction methods for time series, such as SPCI, can only guarantee the asymptotic conditional coverage, rather than the exact finite-sample marginal coverage available for classical conformal prediction under exchangeability.

Despite the empirical success, existing sequential conformal procedures rely heavily on estimating tail conditional quantiles of nonconformity score separately at each quantile level, and compute the prediction interval by directly plugging those tail conditional quantile estimators into the interval endpoints. *This design has two practical drawbacks that become pronounced for time series.* First, tail quantile estimators are intrinsically noisy and can fluctuate substantially across time, leading to volatile interval widths. Second, because different quantile levels are treated separately, the resulting estimated quantile curve may fail to be mono-

---

[1]Booth School of Business, University of Chicago, Chicago, USA [2]Wang Yanan Institute for Studies in Economics, Xiamen University, Xiamen, China [3]Department of Finance, School of Economics, Xiamen University, Xiamen, China [4]Department of Statistics and Actuarial Science, School of Computing and Data Science, The University of Hong Kong, Hong Kong. Correspondence to: Zhoufan Zhu <tylerzzf@xmu.edu.cn>.

*Proceedings of the 43$^{rd}$ International Conference on Machine Learning*, Seoul, South Korea. PMLR 306, 2026. Copyright 2026 by the author(s).

tone, producing quantile crossing. When users evaluate multiple significance levels (e.g., for model monitoring, risk reporting, or downstream decision rules), quantile crossing manifests as non-nested prediction intervals across significance levels, undermining interpretability and complicating deployment.

We address the above issues by introducing *Conditional Quantile Adjusted Conformal Prediction* (CQACP), a sequential conformal inference framework designed to produce stable and nested prediction intervals for time series. The key idea is to treat the conditional quantile function of the nonconformity score as a structured object, rather than a collection of conditional quantile estimators learned separately. Figure 1 compares the prediction intervals produced by SPCI and our CQACP method. The SPCI intervals exhibit quantile crossing, whereas CQACP delivers smooth, well-behaved, and nested prediction intervals.

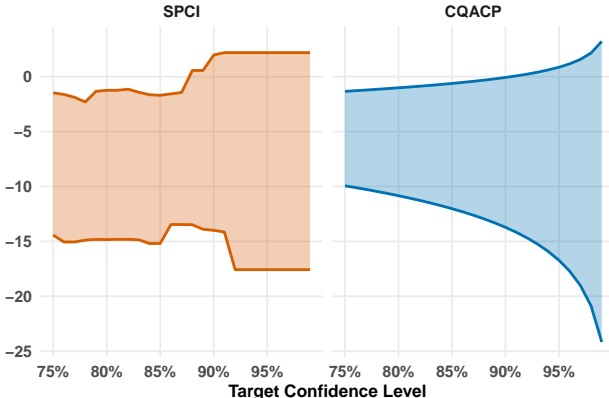

*Figure 1.* Comparison of prediction intervals across confidence level $1 - \alpha$ for SPCI and CQACP, where $\alpha$ is the significance level. SPCI has non-nested intervals, due to the use of quantile autoregression at each quantile level separately, whereas CQACP produces smooth and nested prediction intervals through our conditional quantile adjustment.

At a high level, CQACP estimates the conditional quantile curve of the next nonconformity score over a grid of quantile levels, and then adjusts this curve through a low-dimensional Cornish–Fisher representation. This adjustment shares information across quantile levels to stabilize noisy tail estimators. The resulting curve is used to calibrate sequential prediction intervals, with monotonicity constraints and a shared asymmetry ratio ensuring non-crossing quantiles and nested intervals across confidence levels.

In summary, our main contributions are three-fold:

- Methodologically, we propose CQACP, a sequential conformal prediction framework that calibrates condi-

tional quantile estimators of nonconformity score by fitting a Cornish-Fisher approximation across quantile levels, and enforces monotonicity to prevent quantile crossing.

- Theoretically, we demonstrate that the Cornish–Fisher projection step used by CQACP can improve the estimation accuracy of conditional quantiles under mild conditions. We then establish asymptotic conditional coverage for time series while explicitly allowing for estimation bias in the base conditional quantile estimators

- Empirically, we find that on several real datasets, CQACP yields smoother and typically narrower prediction intervals than various baselines while maintaining target coverage, and produces nested prediction intervals across significance levels for a range of conditional quantile learners.

## 2. Background and Related Work

### 2.1. Conformal prediction and exchangeability

A finite sequence $Z_1, \ldots, Z_n$ is exchangeable if $(Z_1, \ldots, Z_n) \stackrel{d}{=} (Z_{\pi(1)}, \ldots, Z_{\pi(n)})$ for every permutation $\pi$ of $\{1, \ldots, n\}$. Classical conformal prediction calibrates a nonconformity score when the calibration data and future test point are exchangeable, and yields exact finite-sample marginal coverage (Vovk et al., 2005; Lei et al., 2013; Lei & Wasserman, 2014). In regression, a common nonconformity score is the prediction residual $\widehat{\varepsilon}_t = Y_t - \widehat{f}(X_t)$, where $\widehat{f}$ is a black-box point predictor. Split/inductive and jackknife+/CV-style variants make conformal prediction practical while preserving exchangeability-based validity (Lei & Wasserman, 2014; Barber et al., 2021b; Angelopoulos & Bates, 2023). Conformalized quantile regression leverages estimated conditional quantiles before applying the same conformal calibration step to improve efficiency (Romano et al., 2019).

When test points are not exchangeable with the calibration sample, exact finite-sample validity is no longer automatic, and strong forms of conditional coverage are impossible without additional assumptions (Barber et al., 2021a). A leading approach is to restore approximate validity by reweighting calibration scores to account for distribution shift, most notably under covariate shift via importance-weighted conformal quantiles (Tibshirani et al., 2019). More generally, Barber et al. (2023) characterize the coverage gap for arbitrary non-exchangeable data in terms of distributional discrepancies and motivate principled weighting schemes, while Guan (2023) proposes localized methods using similarity weighting to adapt the calibration distribution to the test point.

## 2.2. Sequential score calibration for time series

Time-series data are generally non-exchangeable because temporal dependence and distributional shifts can change the joint distribution under permutations. A common model-agnostic approach recalibrates prediction residuals using rolling or exponentially weighted score windows. Its adaptive variants further update the effective miscoverage level over time to maintain long-run validity under distributional shift; see Adaptive Conformal Inference (ACI; Gibbs & Candès, 2021) and its follow-ups (Zaffran et al., 2022; Gibbs & Candès, 2024; Wu et al., 2025). More recent approaches model the score process directly–for example via ensemble predictors (Xu & Xie, 2021), sequential quantile learning (Xu & Xie, 2023) or kernel-weighted score quantiles with mixing-based analysis (Lee et al., 2025), targeting sharper prediction intervals with approximate conditional guarantees.

For time series, residual scores are generally serially dependent: The conditional distribution of $\widehat{\varepsilon}_t$ may depend on past observations or past scores. For a target miscoverage level $\alpha \in (0, 1)$ and a window length $w$, sequential conformal methods therefore update the score calibration over time. Let $E_t^w = (\widehat{\varepsilon}_{t-1}, \ldots, \widehat{\varepsilon}_{t-w})$ denote the recent residual window, and let $q_p(S)$ denote the empirical $p$-quantile of the scores in a finite set $S$. EnbPI uses recent residuals and empirical score quantiles to form

$$[\widehat{f}(X_t) + q_{\alpha/2}(E_t^w), \widehat{f}(X_t) + q_{1-\alpha/2}(E_t^w)].$$

By contrast, SPCI estimates the conditional score quantile

$$Q_t(p) = \inf\{e : \mathbb{P}(\widehat{\varepsilon}_t \leq e \mid \mathcal{F}_{t-1}) \geq p\}, \text{ for } p \in (0, 1),$$

where $\mathcal{F}_{t-1}$ denotes the information available before predicting $Y_t$ and is formalized in Section 3. Let $\widehat{Q}_t(p)$ be an estimator of $Q_t(p)$ learned from past residual histories. SPCI then constructs asymmetric intervals of the form

$$[\widehat{f}(X_t) + \widehat{Q}_t(\widehat{\beta}_t), \widehat{f}(X_t) + \widehat{Q}_t(1 - \alpha + \widehat{\beta}_t)],$$

where $\widehat{\beta}_t \in [0, \alpha]$ is selected to minimize the estimated interval width. These approaches motivate CQACP, but raw conditional quantile estimators can be noisy in the tails and may cross across quantile levels when estimated separately. CQACP modifies this conditional score-quantile estimation step by treating $p \mapsto Q_t(p)$ as a structured curve.

## 2.3. Cornish–Fisher expansion

The CF expansion (Cornish & Fisher, 1938) approximates a quantile function by a polynomial in the Gaussian quantile $z_p = \Phi^{-1}(p)$, with coefficients determined by distributional moments. In our notation, for a truncation order $K$,

$$Q_t(p) \approx \psi_K(z_p)^\top \theta_{t,K},$$
$$\psi_K(z) = (H_0(z), \ldots, H_{K-1}(z))^\top,$$

where $H_j$ are probabilists' Hermite polynomials. This representation is useful for time-series calibration because the coefficients can vary with $t$, while the low-dimensional basis shares information across quantile levels.

Recent work has used the CF expansion and related quantile-to-moment ideas to recover distributional moments from estimated quantiles, including conditional moment estimation for financial time series (Zhang & Zhu, 2026) and variance control in distributional reinforcement learning (Kuang et al., 2023; Zhu & Zhu, 2025). Our work brings this perspective to sequential conformal calibration by using CF-based cross-quantile information sharing to stabilize conditional score quantiles.

## 3. Problem Setup

Consider a time series of observations $\{(X_t, Y_t), t = 1, 2, \ldots\}$, where $X_t \in \mathbb{R}^d$ is a $d$-dimensional feature vector and $Y_t \in \mathbb{R}$ is a continuous scalar response variable. The components of $X_t$ may include exogenous variables, lagged values of $Y_t$, or both. Let the first $T$ observations $\{(X_t, Y_t)\}_{t=1}^T$ form the in-sample data. Our goal is to construct prediction intervals $\widehat{C}_{\alpha,t-1}(X_t)$ sequentially for $t = T+1, T+2, \ldots$, such that the prediction interval can contain the true response variable $Y_t$ with probability at least $1 - \alpha$. Here, $\widehat{C}_{\alpha,t-1}(X_t)$ depends on a pre-specified significance level $\alpha \in (0, 1)$ and a point prediction $\widehat{Y}_t := \widehat{f}(X_t)$ produced by a prediction model $\widehat{f} : \mathbb{R}^d \to \mathbb{R}$. The subscripts $\alpha$ and $t - 1$ in $\widehat{C}_{\alpha,t-1}(\cdot)$ indicate the target significance level and the information available up to time $t - 1$, respectively.

Here and below, the nonconformity score is the prediction residual $\widehat{\varepsilon}_t = Y_t - \widehat{f}(X_t)$. Its serial dependence means that the conditional distribution of $\widehat{\varepsilon}_t$ may depend on past observations or past scores, so our goal is sequential conditional calibration rather than exchangeability-based finite-sample calibration.

Two types of coverage guarantees are commonly considered for $\widehat{C}_{\alpha,t-1}(X_t)$. The first one is *marginal coverage*:

$$\mathbb{P}\big(Y_t \in \widehat{C}_{\alpha,t-1}(X_t)\big) \geq 1 - \alpha, \tag{1}$$

which ensures that, on average over the data-generating process, the interval contains $Y_t$ at least $1 - \alpha$ of the time. The second one is *conditional coverage*:

$$\mathbb{P}\big(Y_t \in \widehat{C}_{\alpha,t-1}(X_t) \mid \mathcal{F}_{t-1}\big) \geq 1 - \alpha, \tag{2}$$

where $\mathcal{F}_{t-1} = \sigma(\{X_s, Y_s\}_{s \leq t-1}, X_t)$ denotes the information set, including past observations up to time $t - 1$ and the current feature vector at time $t$. Notably, conditional coverage is a stronger requirement than marginal coverage, as it requires the prediction interval to contain $Y_t$ with probability at least $1 - \alpha$ given any realized history.

Beyond coverage guarantees, we also seek prediction intervals that are as short as possible, that is, intervals that minimize the length $\left|\widehat{C}_{\alpha,t-1}(X_t)\right|$, in order to improve informativeness and practical usefulness. Particularly, in financial risk management, overly wide intervals imply excessively conservative capital buffers and reduced capital efficiency. Meanwhile, in energy load forecasting, shorter reliable intervals support more precise reserve scheduling and lower operating costs. Hence, minimizing the length $\left|\widehat{C}_{\alpha,t-1}(X_t)\right|$ is practically important.

In addition, we require the *nested prediction set* property (Szabadváry & Löfström, 2026): for any $0 < \alpha_1 < \alpha_2 < 1$ and any feature value $X_t$,

$$\widehat{C}_{\alpha_2,t-1}(X_t) \subseteq \widehat{C}_{\alpha_1,t-1}(X_t), \tag{3}$$

so that the intervals expand monotonically as the significance level decreases (i.e., confidence level increases). This property is crucial in practice, as it ensures coherent uncertainty quantification: higher confidence levels always lead to wider and more conservative prediction intervals. This avoids counterintuitive crossings across confidence levels and supports reliable decision-making and risk assessment. See Figure 1 for both negative and positive examples.

# 4. Conditional Quantile Adjusted Conformal Prediction

We now present CQACP. The background above motivates our focus on the conditional quantile curve of the next score. Rather than using the raw estimators $\widehat{Q}_t(p)$ separately at each quantile level, CQACP projects the full estimated curve onto a low-dimensional CF/Hermite space, enforces monotonicity, and then constructs nested asymmetric sequential intervals.

## 4.1. CF adjustment of the conditional score-quantile curve

CQACP starts from a base conditional quantile learner for the residual score $\widehat{\varepsilon}_t$, and then pools the resulting grid-level estimators through the CF representation described in Section 2.3.

Using the notation from Section 2.3, we write the order-$K$ CF approximation to the true conditional score quantile as

$$Q_t(p) = \psi_K(z_p)^\top \theta_{t,K} + R_{K,t}(p), \tag{4}$$

where $\theta_{t,K} := \left(\theta_t^{(1)}, \theta_t^{(2)}, \ldots, \theta_t^{(K)}\right)^\top \in \mathbb{R}^K$ is a (time-varying) coefficient vector depending on conditional moments of $\widehat{\varepsilon}_t$, and $R_{K,t}(p)$ collects contributions from higher-order terms beyond the truncation order. We treat $R_{K,t}(p)$ as the truncation error and estimate the finite-dimensional coefficient vector directly from the raw conditional quantile curve.

Fix a grid of quantile levels $\mathcal{A} = \{p_1, \ldots, p_n\} \subset (0,1)$ (e.g., $\mathcal{A} = \{0.01, 0.02, \ldots, 0.99\}$). At time $t$, we apply a base conditional quantile learner on this grid to obtain the vector of estimated conditional quantiles $\widehat{q}_{t,i} := \widehat{Q}_t(p_i)$, $i = 1, \ldots, n$. We treat these base conditional quantile estimators $\{\widehat{q}_{t,i}\}_{i=1}^n$ as noisy and possibly biased inputs and then denoise them by fitting a CF approximation across many quantile levels. This shrinks unstable base tail conditional quantile estimators toward the global shape implied by the full quantile curve. Specifically, motivated by (4), we estimate $\theta_{t,K}$ by fitting a CF approximation to the set $\{\widehat{q}_{t,i}\}_{i=1}^n$:

$$\min_{\theta \in \mathbb{R}^K} \sum_{i=1}^n \left[\widehat{q}_{t,i} - \psi_K(z_{p_i})^\top \theta\right]^2. \tag{5}$$

Let $Z_K \in \mathbb{R}^{n \times K}$ be the design matrix with the $i$-th row $(Z_K)_{i\cdot} = \psi_K(z_{p_i})^\top$. Solving (5) leads to the ordinary least squares (OLS) estimator of $\theta$:

$$\widehat{\theta}_{t,K}^{\text{OLS}} := (Z_K^\top Z_K)^{-1} Z_K^\top \widehat{\mathbf{q}}_t, \tag{6}$$

where $\widehat{\mathbf{q}}_t := (\widehat{q}_{t,1}, \widehat{q}_{t,2}, \ldots, \widehat{q}_{t,n})^\top$.

We then define the adjusted conditional quantile curve

$$\widehat{Q}_{t,K}^{\text{adj}}(p) = \psi_K(z_p)^\top \widehat{\theta}_{t,K}, \tag{7}$$

where $\widehat{\theta}_{t,K}$ is chosen as $\widehat{\theta}_{t,K}^{\text{OLS}}$ if it yields a monotone curve, and otherwise replaced by a constrained least squares (CLS) estimator as described next. Since $\widehat{Q}_{t,K}^{\text{adj}}(p)$ is obtained by pooling information across many quantile levels through a CF approximation, it yields a smooth quantile curve in $p$ and stabilizes tail behavior relative to a separate quantile learning at each individual quantile level.

## 4.2. Monotonicity and nested interval construction

Although (7) is smooth in $p$, finite-sample noise can lead to a non-monotone curve (i.e., quantile crossing). Considering that $p \mapsto z_p$ is strictly increasing, a sufficient condition for non-crossing on the probability range of interest is that the curve is non-decreasing on the $z$-scale:

$$\frac{\partial}{\partial z} \widehat{Q}_{t,K}^{\text{adj}}(\Phi(z)) = \left(\psi_K'(z)\right)^\top \widehat{\theta}_{t,K} \geq 0, \quad \forall z \in \mathcal{Z}, \tag{8}$$

where $\psi_K'(z)$ is the element-wise derivative of $\psi_K(z)$ and $\mathcal{Z} := \{z_{p_i} : p_i \in \mathcal{A}\}$. In practice, we first compute the unconstrained OLS solution in (6) and check whether the derivative condition in (8) holds over $\mathcal{Z}$. If any violation is detected at certain time point $t$, we re-estimate $\theta$ using the CLS loss:

$$\widehat{\theta}_{t,K}^{\text{CLS}} = \arg\min_{\theta \in \mathbb{R}^K} \sum_{i=1}^n \left[\widehat{q}_{t,i} - \psi_K(z_{p_i})^\top \theta\right]^2,$$
$$\text{s.t.} \quad \left(\psi_K'(z)\right)^\top \theta \geq 0, \quad \forall z \in \mathcal{Z}. \tag{9}$$

The resulting estimator $\widehat{\theta}_{t,K}^{\mathrm{CLS}} := (\widehat{\theta}_{1,t}, \widehat{\theta}_{2,t}, \ldots, \widehat{\theta}_{K,t})^\top$ is then used in (7) at this time point $t$. Clearly, the monotonicity constraint ensures a non-crossing adjusted conditional quantile curve in $p$ and, consequently, guarantees nesting of the resulting prediction intervals.

Besides replacing $\widehat{Q}_t$ with $\widehat{Q}_{t,K}^{\mathrm{adj}}$, we retain the width-minimizing asymmetry parameter from SPCI, but fix it across significance levels to ensure that the resulting prediction intervals are nested. Specifically, we introduce a time-varying asymmetry ratio $\widehat{\rho}_t \in (0,1)$ and define

$$\widehat{\rho}_t = \arg\min_{\rho \in (0,1)} \left\{ \widehat{Q}_{t,K}^{\mathrm{adj}}(1 - \alpha_0 + \rho\alpha_0) \right.$$
$$\left. - \widehat{Q}_{t,K}^{\mathrm{adj}}(\rho\alpha_0) \right\}, \quad (10)$$

where $\alpha_0$ is a reference significance level[1].

After determining $\widehat{\rho}_t$ given $\alpha_0$, for any $\alpha \in (0,1)$, the one-step-ahead prediction interval is then

$$\widehat{C}_{\alpha,t-1}(X_t) = \left[ \widehat{f}(X_t) + \widehat{Q}_{t,K}^{\mathrm{adj}}(\widehat{\rho}_t\alpha), \right.$$
$$\left. \widehat{f}(X_t) + \widehat{Q}_{t,K}^{\mathrm{adj}}(1 - \alpha + \widehat{\rho}_t\alpha) \right]. \quad (11)$$

Since $\widehat{Q}_{t,K}^{\mathrm{adj}}(p)$ is non-decreasing in $p$, this construction guarantees nesting: if $\alpha_1 \geq \alpha_2$, then $\widehat{C}_{\alpha_1,t-1}(X_t) \subseteq \widehat{C}_{\alpha_2,t-1}(X_t)$. Overall, Algorithm 1 presents the practical implementation of CQACP.

### 4.3. Implementation and selection of $K$

Intuitively, the truncation order $K$ reflects a bias-variance trade-off. A larger $K$ can reduce CF approximation bias by incorporating higher-order conditional moments, but it also tends to increase estimation variance because such moments are noisy to estimate. Since the optimal $K$ is difficult to justify theoretically, we select it using cross-validation.

Specifically, given a grid of quantile levels $\mathcal{A}$ and the corresponding conditional quantile estimators $\widehat{q}_{t,i}$ for $p_i \in \mathcal{A}$ at time $t$ over a tuning index set $\mathcal{T}_{\mathrm{tune}}$, we split $\mathcal{A}$ into $J$ folds $\{\mathcal{A}_j\}_{j=1}^J$. For each candidate $K \in \mathcal{K}$ and each fold $j$, we fit a CF approximation using quantile levels in $\mathcal{A} \setminus \mathcal{A}_j$ and obtain an adjusted quantile curve $\widehat{Q}_{t,K,-j}^{\mathrm{adj}}(\cdot)$. We evaluate $K$ using the average $L_2$ discrepancy on the held-out levels,

$$\mathrm{CV}(K) = \frac{1}{J}\sum_{j=1}^J \frac{1}{|\mathcal{A}_j|} \sum_{p \in \mathcal{A}_j} \left[ \widehat{Q}_{t,K,-j}^{\mathrm{adj}}(p) - \widehat{Q}_t(p) \right]^2. \quad (12)$$

In practice, we set $\widehat{K} = \arg\min_{K \in \mathcal{K}} \mathrm{CV}(K)$, and plug in $\widehat{K}$ for $\widehat{Q}^{\mathrm{adj}}$ in (7).

---

[1] A common choice for $\alpha_0$ is the target significance level.

---

**Algorithm 1** CQACP

**Input:** training sample $\{(X_t, Y_t)\}_{t=1}^T$; testing features $\{X_t\}_{t=T+1}^{T^*}$; window length $w$; significance level $\alpha \in (0,1)$; reference level for nesting $\alpha_0 \in (0,1)$; quantile grid $\mathcal{A} = \{p_1, \ldots, p_n\} \subset (0,1)$; CF truncation order $K \geq 3$; pre-trained point predictor $\widehat{f}$.

**Output:** Prediction intervals $\{\widehat{C}_{\alpha,t-1}(X_t)\}_{t=T+1}^{T^*}$.

1: **# Initialization.**
2: Compute in-sample residuals $\widehat{\varepsilon}_s \leftarrow Y_s - \widehat{f}(X_s)$ for $s = 1, \ldots, T$.
3: **for** $j = w+1, \ldots, T$ **do**
4: $\quad E_j^w \leftarrow (\widehat{\varepsilon}_{j-1}, \ldots, \widehat{\varepsilon}_{j-w})$.
5: **end for**
6: $\mathcal{D}_T \leftarrow \{(E_j^w, \widehat{\varepsilon}_j) : j = w+1, \ldots, T\}$.
7: **for** $t = T+1, \ldots, T^*$ **do**
8: $\quad$ **# Quantile predictions.**
9: $\quad E_t^w \leftarrow (\widehat{\varepsilon}_{t-1}, \ldots, \widehat{\varepsilon}_{t-w})$ using only residuals observed before time $t$.
10: $\quad$ Obtain the quantile estimator $\widehat{Q}_t(\cdot)$ from $\mathcal{D}_{t-1}$ at the current residual history $E_t^w$.
11: $\quad \widehat{q}_{t,i} \leftarrow \widehat{Q}_t(p_i), z_i \leftarrow \Phi^{-1}(p_i)$ for $i = 1, \ldots, n$.
$\quad$ **# Quantile adjustment.**
12: $\quad$ Obtain $\widehat{\theta}_{t,K}$ via (6), or via (9) if the OLS curve violates monotonicity.
13: $\quad$ Define $\widehat{Q}_{t,K}^{\mathrm{adj}}(p) \leftarrow \psi_K(\Phi^{-1}(p))^\top \widehat{\theta}_{t,K}$ for all $p \in (0,1)$.
14: $\quad$ **# Nested asymmetry selection.**
15: $\quad \widehat{\rho}_t \leftarrow \arg\min_{\rho \in (0,1)} \{ \widehat{Q}_{t,K}^{\mathrm{adj}}(1 - \alpha_0 + \rho\alpha_0) - \widehat{Q}_{t,K}^{\mathrm{adj}}(\rho\alpha_0) \}$.
16: $\quad$ **# Prediction interval construction.**
17: $\quad \widehat{C}_{\alpha,t-1}(X_t) \leftarrow \left[ \widehat{f}(X_t) + \widehat{Q}_{t,K}^{\mathrm{adj}}(\widehat{\rho}_t\alpha), \widehat{f}(X_t) + \widehat{Q}_{t,K}^{\mathrm{adj}}(1 - \alpha + \widehat{\rho}_t\alpha) \right]$.
18: $\quad$ **# Rolling update.**
19: $\quad$ After observing $Y_t$, compute $\widehat{\varepsilon}_t \leftarrow Y_t - \widehat{f}(X_t)$.
20: $\quad E_{t+1}^w \leftarrow (\widehat{\varepsilon}_t, \widehat{\varepsilon}_{t-1}, \ldots, \widehat{\varepsilon}_{t-w+1})$.
21: $\quad \mathcal{D}_t \leftarrow \mathcal{D}_{t-1} \cup \{(E_t^w, \widehat{\varepsilon}_t)\}$.
22: **end for**

---

## 5. Theoretical Analysis

In this section, we first show that in terms of mean squared error (MSE), our adjusted conditional quantile estimator is more accurate than the base conditional quantile estimator. We then establish the asymptotic coverage of our method under the settings where the base conditional quantile estimator can be biased and the nonconformity scores have serial dependence.

The assumptions used in Theorems 5.1–5.2 below are standard regularity conditions from quantile regression and sieve estimation, rather than restrictions to a parametric time-series model. For the MSE improvement result in Theorem

5.1, we require the quantile grid to stay in a central probability region, the transformed conditional score-quantile curve $z \mapsto Q_t(\Phi(z))$ to be smooth, and the CF truncation order $K$ to grow slowly relative to the grid size $n$; we also allow the base learner to return biased and noisy grid-level quantile estimates. For the conditional coverage result in Theorem 5.2, we additionally assume local regularity of the conditional distribution of the next score, a well-conditioned CF/Hermite design, consistency of the CF-projected component of the base quantile curve, and endpoint probabilities that remain in the same central region. These are high-level conditions on the learned score-quantile curve and its CF projection. They allow serial dependence through the conditioning information set $\mathcal{F}_{t-1}$, and most importantly, they do not impose exchangeability or parametric dynamics for the time series. Full assumptions and proofs are in Appendix A.

Denote the true conditional quantile vector and the adjusted conditional quantile estimator vector on $\mathcal{A}$, respectively, as

$$\mathbf{q}_t := \big(Q_t(p_1), \ldots, Q_t(p_n)\big)^\top \text{ and}$$
$$\widehat{\mathbf{q}}_{t,K}^{\mathrm{adj}} := \big(\widehat{Q}_{t,K}^{\mathrm{adj}}(p_1), \ldots, \widehat{Q}_{t,K}^{\mathrm{adj}}(p_n)\big)^\top.$$

**Theorem 5.1** (MSE improvement). *Under Assumptions 1–4, there exists a constant $C > 0$ such that*

$$\mathbb{E}\Big[\|\widehat{\mathbf{q}}_t - \mathbf{q}_t\|_n^2 - \|\widehat{\mathbf{q}}_{t,K}^{\mathrm{adj}} - \mathbf{q}_t\|_n^2 \,\Big|\, \mathcal{F}_{t-1}\Big] \geq$$
$$\sigma^2\Big(1 - \frac{K}{n}\Big) \; - \; CK^{-2s},$$

*where $\|v\|_n^2 := \frac{1}{n}\|v\|_2^2$ denotes the empirical norm, $s > 0$ is the Hölder smoothness exponent of the transformed quantile curve $z \mapsto Q_t(\Phi(z))$ (as in Assumption 2), and $\sigma^2 > 0$ is the constant satisfying $\mathrm{Cov}(\widehat{\mathbf{q}}_t - \mathbf{q}_t \mid \mathcal{F}_{t-1}) \succeq \sigma^2 I_n$ (as in Assumption 3). In particular, for sufficiently large $n$,*

$$\mathbb{E}\Big[\|\widehat{\mathbf{q}}_{t,K}^{\mathrm{adj}} - \mathbf{q}_t\|_n^2 \,\Big|\, \mathcal{F}_{t-1}\Big] < \mathbb{E}\big[\|\widehat{\mathbf{q}}_t - \mathbf{q}_t\|_n^2 \,\big|\, \mathcal{F}_{t-1}\big].$$

Theorem 5.1 provides an explicit lower bound on the conditional MSE gain from CQACP:

$$\text{MSE gain} \gtrsim \underbrace{\sigma^2\Big(1 - \frac{K}{n}\Big)}_{\text{variance removed by projection}} - \underbrace{CK^{-2s}}_{\text{approximation penalty}}.$$

The first term quantifies how much variability across quantile levels is removed when projecting the base quantile estimator vector onto an order-$K$ CF approximation, while the second term is the price paid for restricting the true quantile curve to the order-$K$ class. Under the growth conditions in Assumption 4, the truncation penalty vanishes faster than the removable variability, implying that the adjusted conditional quantile estimator strictly improves the grid-level MSE for all sufficiently large $n$.

Notably, Theorem 5.1 allows the base conditional quantile estimator $\widehat{Q}_t(p)$ to be biased. As shown in Lemma A.7 in Appendix A, our adjusted conditional quantile estimator $\widehat{Q}_{t,K}^{\mathrm{adj}}(p)$ can remove the impact of bias from $\widehat{Q}_t(p)$ to achieve consistency. In other words, using the idea of CF approximation, $\widehat{Q}_{t,K}^{\mathrm{adj}}(p)$ accomplishes de-noising in learning quantiles, while, simultaneously, avoiding quantile crossing. Hence, $\widehat{Q}_{t,K}^{\mathrm{adj}}(p)$ can have a large application scope beyond the current task of conformal prediction.

**Theorem 5.2** (Asymptotic conditional coverage). *Under Assumptions 1–2 and 4–9, as the sample size $T \to \infty$, we have asymptotic $(1 - \alpha)$ conditional coverage for the prediction interval in (11) at $t = T + 1$:*

$$\big|\mathbb{P}\big(Y_t \in \widehat{C}_{\alpha,t-1}(X_t) \mid \mathcal{F}_{t-1}\big) - (1 - \alpha)\big| \xrightarrow{\mathbb{P}} 0.$$

Theorem 5.2 formalizes the validity guarantee of CQACP in the time series setting: Although finite-sample conditional coverage is generally impossible under dependence without strong assumptions, CQACP achieves asymptotic $(1 - \alpha)$ coverage due to the consistency of $\widehat{Q}_{t,K}^{\mathrm{adj}}(p)$.

# 6. Experiments

Beyond theoretical analysis, we benchmark CQACP against existing sequential conformal prediction methods on real-world datasets. The baseline methods most closely related to CQACP are EnbPI (Xu & Xie, 2021) and SPCI (Xu & Xie, 2023), both of which perform conformal inference by dynamically updating the residual distribution.

Another class of online conformal methods targets long-run coverage by adaptively adjusting the deployed miscoverage level over time, including ACI (Gibbs & Candès, 2021), Online (Sub)Gradient Descent and its scale-free variant (OGD and SF-OGD; Bhatnagar et al., 2023), conformal PID control (PID; Angelopoulos et al., 2023), and Error-quantified Conformal Inference (ECI; Wu et al., 2025).

In addition to these post-training approaches, we also consider probabilistic forecasting models combined with conformal prediction as competitors, including DeepAR (Salinas et al., 2020) and Temporal Fusion Transformer (TFT; Lim et al., 2021).

We evaluate all methods on four real-world datasets covering financial volatility, asset returns, foreign exchange rates, and electricity demand: *Realized Volatility*, *Index Return*, *Exchange Rates*, and *Electricity*. More dataset descriptions and implementation details are provided in Appendices D.1 and D.2, respectively. *The source code for reproducing the experiments is provided in the supplementary material.*

*Table 1.* Empirical coverage and average width of prediction intervals across different methods and real-world time series with $\alpha = 10\%$. Ideally, the coverage should be close to the nominal confidence level $1 - \alpha = 90\%$, while the average interval width should be as small as possible. The standard deviation of each method over five independent runs is reported in parentheses, where "-" indicates zero standard deviation. Results for the CQACP method are highlighted in bold.

| Method | Realized Volatility | | | | Index Return | | | | Exchange Rates | | | | Electricity | | | |
|---|---|---|---|---|---|---|---|---|---|---|---|---|---|---|---|---|
| | Coverage | | Width | | Coverage | | Width | | Coverage | | Width | | Coverage | | Width | |
| EnbPI | 91.2% | (0.4%) | 2.7% | (4.0%) | 88.8% | (1.0%) | 3.7% | (0.0%) | 89.1% | (0.4%) | 0.6% | (0.0%) | 85.9% | (0.4%) | 16.9% | (0.0%) |
| SPCI | 89.0% | (0.5%) | 0.8% | (0.0%) | 87.9% | (0.3%) | 3.4% | (0.0%) | 89.3% | (0.2%) | 0.4% | (0.0%) | 88.9% | (0.3%) | 9.3% | (0.0%) |
| **CQACP** | **90.1%** | **(0.2%)** | **0.9%** | **(0.0%)** | **90.0%** | **(0.2%)** | **3.4%** | **(0.0%)** | **90.1%** | **(0.3%)** | **0.4%** | **(0.0%)** | **90.0%** | **(0.1%)** | **9.2%** | **(0.0%)** |
| ACI | 89.9% | - | $\infty$ | - | 90.1% | - | $\infty$ | - | 90.0% | - | $\infty$ | - | 90.0% | - | $\infty$ | - |
| OGD | 89.9% | - | 44.9% | - | 89.9% | - | 44.9% | - | 90.0% | - | 45.0% | - | 90.0% | - | 48.3% | - |
| SF-OGD | 89.9% | - | 4.3% | - | 89.9% | - | 5.9% | - | 90.0% | - | 3.2% | - | 90.0% | - | 16.2% | - |
| PID | 90.1% | - | 2.6% | - | 90.1% | - | 7.0% | - | 90.2% | - | 1.4% | - | 90.0% | - | 25.0% | - |
| ECI | 89.9% | - | 2.3% | - | 89.9% | - | 6.6% | - | 90.0% | - | 1.3% | - | 90.0% | - | 24.1% | - |
| DeepAR | 89.9% | (2.9%) | 0.9% | (0.1%) | 85.3% | (1.0%) | 3.1% | (0.1%) | 89.3% | (0.6%) | 0.9% | (0.0%) | 93.0% | (1.0%) | 16.7% | (1.0%) |
| TFT | 92.3% | (5.7%) | 1.1% | (0.2%) | 88.4% | (1.0%) | 3.3% | (0.1%) | 87.9% | (0.5%) | 0.8% | (0.0%) | 91.6% | (1.1%) | 21.3% | (1.3%) |

## 6.1. Comparison across different methods

Table 1 reports the coverage and width of prediction intervals across different methods and real-world datasets. From this table, we have the following findings.

First, compared to the closest baselines, EnbPI and SPCI, the proposed CQACP method delivers more reliable and robust coverage across all considered datasets. While the closest-to-nominal coverage among EnbPI and SPCI is 89.0%, 88.8%, 89.3%, and 88.9%, CQACP attains empirical coverages of 90.1%, 90.0%, 90.1%, and 90.0%, which are closer to the nominal confidence level. Moreover, CQACP exhibits small variability across trials, with standard deviations of 0.2%, 0.2%, 0.3%, and 0.1%, outperforming EnbPI and SPCI on nearly all datasets, except for Exchange Rates.

Second, compared to ACI-type methods, CQACP achieves comparable empirical coverage with substantially narrower prediction intervals. ACI-type methods target long-run coverage by adaptively updating the deployed miscoverage level. Empirically, they often achieve coverage close to the target, but at the cost of substantially wider, and occasionally unbounded intervals. Their narrowest interval widths are 2.3%, 6.6%, 1.3%, and 24.1% across the four datasets. In contrast, CQACP reduces the interval widths to 0.9%, 3.4%, 0.4%, and 9.2%, which are less than half of those produced by ACI-type methods. As a result, CQACP provides tight and practically useful prediction intervals without compromising coverage accuracy.

Third, the probabilistic forecasting models, DeepAR and TFT, exhibit less consistent performance across datasets. Although they occasionally achieve competitive interval widths, their coverage is often unstable and deviates from the nominal level, especially in the energy domain. This suggests that purely probabilistic models may struggle to provide reliable uncertainty quantification without explicit

conformal calibration.

## 6.2. Impact of significance level

Due to space limitations, we only report the results of CQACP, EnbPI, and SPCI in this subsection. Results for the remaining methods can be found in Appendix E.

As shown in Table 2, across all choices of significance level $\alpha$, CQACP consistently attains empirical coverage closest to the nominal confidence level $1 - \alpha$, while maintaining interval widths that are comparable to or smaller than those of EnbPI and SPCI. In addition, CQACP shows smaller variability across multiple random trials, indicating improved stability. Overall, the nested prediction intervals produced by CQACP consistently outperform those of EnbPI and SPCI by a substantial margin, regardless of the choice of significance level.

## 6.3. Impact of base quantile learner

To further examine the robustness of CQACP to quantile estimation errors, we conduct an ablation study using three base quantile learners: QRF (Shiraishi et al., 2024), quantile autoregression (QAR; Koenker & Xiao, 2006), and $K$-nearest-neighbor quantile regression (KNNQR; Ma et al., 2016). These models differ markedly in stability and approximation accuracy, enabling us to assess the sensitivity of sequential conformal methods to the quality of the underlying quantile estimators. Since EnbPI and the other baselines do not rely on a quantile learner, we benchmark CQACP only against SPCI.

Table 3 reports the empirical coverage and interval width across different base quantile learners. The results show that CQACP consistently achieves empirical coverage closer to the nominal level than SPCI among all settings, while SPCI exhibits noticeable undercoverage when weaker or

*Table 2.* Empirical coverage and average width of prediction intervals across different choices of significance level $\alpha$. Ideally, the coverage should be close to the nominal confidence level $1 - \alpha$, while the average interval width should be as small as possible. The standard deviation of each method over five independent runs is reported in parentheses, where "-" indicates zero standard deviation. Results for the CQACP method are highlighted in bold.

| | Realized Volatility | | Index Return | | Exchange Rates | | Electricity | |
|---|---|---|---|---|---|---|---|---|
| Method | Coverage | Width | Coverage | Width | Coverage | Width | Coverage | Width |
| | | | Panel A: $\alpha = 10\%$ | | | | | |
| EnbPI | 91.2% (0.4%) | 2.7% (4.0%) | 88.8% (1.0%) | 3.7% (0.0%) | 89.1% (0.4%) | 0.6% (0.0%) | 85.9% (0.4%) | 16.9% (0.0%) |
| SPCI | 89.0% (0.5%) | 0.8% (0.0%) | 87.9% (0.3%) | 3.4% (0.0%) | 89.3% (0.2%) | 0.4% (0.0%) | 88.9% (0.3%) | 9.3% (0.0%) |
| **CQACP** | **90.1% (0.2%)** | **0.9% (0.0%)** | **90.0% (0.2%)** | **3.4% (0.0%)** | **90.1% (0.3%)** | **0.4% (0.0%)** | **90.0% (0.1%)** | **9.2% (0.0%)** |
| | | | Panel B: $\alpha = 5\%$ | | | | | |
| EnbPI | 96.1% (0.2%) | 1.3% (0.0%) | 94.5% (0.8%) | 4.8% (0.0%) | 94.3% (0.2%) | 0.9% (0.0%) | 91.3% (0.5%) | 21.6% (0.1%) |
| SPCI | 93.4% (0.3%) | 1.1% (0.0%) | 93.3% (0.8%) | 4.2% (0.1%) | 94.1% (0.2%) | 0.5% (0.0%) | 93.4% (0.3%) | 10.7% (0.1%) |
| **CQACP** | **95.0% (0.2%)** | **1.2% (0.0%)** | **95.0% (0.1%)** | **4.3% (0.0%)** | **95.0% (0.1%)** | **0.5% (0.0%)** | **95.0% (0.1%)** | **10.6% (0.1%)** |
| | | | Panel C: $\alpha = 2.5\%$ | | | | | |
| EnbPI | 98.2% (0.2%) | 1.8% (0.0%) | 97.2% (0.4%) | 5.8% (0.0%) | 97.2% (0.1%) | 1.2% (0.0%) | 94.4% (0.1%) | 25.6% (0.1%) |
| SPCI | 96.0% (0.3%) | 1.4% (0.0%) | 95.9% (0.5%) | 4.9% (0.0%) | 96.4% (0.3%) | 0.6% (0.0%) | 95.8% (0.4%) | 12.1% (0.1%) |
| **CQACP** | **97.5% (0.1%)** | **1.4% (0.0%)** | **97.5% (0.1%)** | **5.1% (0.0%)** | **97.5% (0.1%)** | **0.7% (0.0%)** | **97.4% (0.2%)** | **12.2% (0.1%)** |

*Table 3.* Empirical coverage and average width of prediction intervals under different base quantile learners with $\alpha = 10\%$. QRF denotes quantile random forests, QAR denotes quantile autoregression, and KNNQR denotes $K$-nearest-neighbor quantile regression. Ideally, empirical coverage should be close to the nominal confidence level $1 - \alpha$, while average interval width should be as small as possible. The standard deviation over five independent runs is reported in parentheses. Results for the CQACP method are highlighted in bold.

| | Realized Volatility | | Index Return | | Exchange Rates | | Electricity | |
|---|---|---|---|---|---|---|---|---|
| Method | Coverage | Width | Coverage | Width | Coverage | Width | Coverage | Width |
| | | | Panel A: QRF | | | | | |
| SPCI | 89.0% (0.5%) | 0.8% (0.0%) | 87.9% (0.3%) | 3.4% (0.0%) | 89.3% (0.2%) | 0.4% (0.0%) | 88.9% (0.3%) | 9.3% (0.0%) |
| **CQACP** | **90.1% (0.2%)** | **0.9% (0.0%)** | **90.0% (0.2%)** | **3.4% (0.0%)** | **90.1% (0.3%)** | **0.4% (0.0%)** | **90.0% (0.1%)** | **9.2% (0.0%)** |
| | | | Panel B: QAR | | | | | |
| SPCI | 87.4% (1.3%) | 0.8% (0.0%) | 88.4% (0.4%) | 3.5% (0.0%) | 90.1% (1.6%) | 0.5% (0.0%) | 85.7% (1.6%) | 7.1% (0.1%) |
| **CQACP** | **89.9% (0.4%)** | **0.8% (0.0%)** | **90.0% (0.2%)** | **3.5% (0.0%)** | **90.0% (0.6%)** | **0.5% (0.0%)** | **89.7% (0.9%)** | **7.4% (0.1%)** |
| | | | Panel C: KNNQR | | | | | |
| SPCI | 87.9% (0.4%) | 0.8% (0.0%) | 88.7% (0.3%) | 3.4% (0.0%) | 88.7% (0.4%) | 0.4% (0.0%) | 87.5% (0.5%) | 12.5% (0.1%) |
| **CQACP** | **89.8% (0.2%)** | **0.8% (0.0%)** | **89.9% (0.1%)** | **3.4% (0.0%)** | **89.5% (0.4%)** | **0.4% (0.0%)** | **90.0% (0.2%)** | **12.4% (0.1%)** |

less stable quantile estimators are used, particularly under QAR and KNNQR. This finding indicates that SPCI is more sensitive to quantile estimation errors.

Moreover, CQACP attains this improvement in coverage without inflating prediction intervals. Its interval widths are comparable to, and often indistinguishable from, those of SPCI, showing that the coverage gains stem from more efficient use of available information rather than increased conservatism.

Taken together, the above findings highlight a key practical advantage of our CQACP method: it does not require uniform convergence of the base quantile learner and can accommodate non-negligible errors in the estimated conditional quantiles.

# 7. Conclusion

In this work, we propose CQACP, a sequential conformal inference framework for time series, which constructs valid and efficient prediction intervals by refining the conditional quantile curve of nonconformity score. Specifically, our CQACP method fits a CF approximation to aggregate information across multiple quantile levels, and then produces a smooth and tail-stable conditional quantile curve. This design yields nested, non-crossing prediction intervals across significance levels while preserving asymptotic conditional validity under serial dependence. Extensive experiments on real-world time series further demonstrate that CQACP consistently achieves state-of-the-art performance compared with existing sequential conformal prediction methods in both coverage and interval width.

Several extensions are worth exploring. First, multi-step forecasting requires joint coverage control across horizons, calling for horizon-aware score definitions and calibration strategies. Second, extending CQACP to multivariate outputs could require jointly calibrated prediction regions that account for cross-component dependence and preserve efficiency (Xu et al., 2024). Third, incorporating semantic feature representations and feature-space weighting (Chen et al., 2024) may allow CQACP to adapt calibration to regime changes by emphasizing relevant historical contexts. Another natural extension is to combine CQACP with ACI-style online level updates, using CQACP as the nested score-quantile engine while adapting the deployed level $\alpha_t$ to target long-run coverage under distribution shifts.

## Acknowledgements

The authors are grateful to the reviewers, Area Chairs, and Program Committee members for their constructive comments and suggestions, which have significantly improved the paper. Zhoufan Zhu's work was supported by the National Natural Science Foundation of China (nos. 72503203 and 72573142), the Natural Science Foundation of Fujian Province (nos. 2025J08009 and 2025J01036), and the Fundamental Research Funds for the Central Universities (no. 20720251053). Ke Zhu's work was supported by the GRF, RGC of Hong Kong (nos. 17302424, 17304723, and 17312622).

## Impact Statement

This paper presents work whose goal is to advance the field of Machine Learning. There are many potential societal consequences of our work, none of which we feel must be specifically highlighted here.

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

# A. Proof

In this appendix, we provide the proof for all theoretical results in Section 5.

## A.1. Assumptions

In the following, Assumptions 1–4 are used for Theorem 5.1, and Assumptions 5–9 are used for Theorem 5.2.

**Assumption 1** (Central grid). The quantile levels $p_i \in [\underline{p}, 1 - \underline{p}]$ for some fixed $\underline{p} \in (0, 1/2)$.

Set $M := \Phi^{-1}(1 - \underline{p})$. Assumption 1 restricts the quantile grid to a central probability range. Equivalently, it ensures that the Gaussian quantile transform $z = \Phi^{-1}(p)$ is uniformly bounded: $z \in [-M, M]$ for $p \in \mathcal{A}$. This is important because the CF-based adjustment fits a polynomial in $z$; allowing $p \to 0$ or 1 would make $|z|$ arbitrarily large and can destabilize both approximation and estimation. In practice, $\underline{p}$ can be chosen small (e.g., 0.01), and the conformal endpoint probabilities used in the final interval are designed to fall inside $[\underline{p}, 1 - \underline{p}]$.

**Assumption 2** (Smoothness). The transformed quantile curve $g_t(z) := Q_t(\Phi(z))$ is Hölder($s$) on $[-M, M]$ for some constant $s > 0$.

Assumption 2 is imposed on the transformed curve $g_t(z)$ over the compact domain $[-M, M]$. Hölder($s$) regularity is a standard condition for polynomial and sieve methods and is used to control the truncation bias incurred by projecting the true quantile curve onto an order-$K$ polynomial class in $z$. In particular, it implies a bound of the form $\|(I - P_K)\mathbf{q}_t\|_n^2 \lesssim K^{-2s}$, which is the key ingredient that allows the truncation error to vanish as $K \to \infty$. Intuitively, smoother conditional quantile curves (i.e., larger $s$) admit faster approximation rates.

**Assumption 3** (Base conditional quantile estimation errors; bias allowed). There exists a constant $\sigma^2 > 0$ (possibly depending on $T$) such that

$$\widehat{\mathbf{q}}_t - \mathbf{q}_t = \mathbf{b}_t + \boldsymbol{\eta}_t, \qquad \mathbb{E}[\boldsymbol{\eta}_t \mid \mathcal{F}_{t-1}] = 0, \qquad \mathrm{Cov}(\boldsymbol{\eta}_t \mid \mathcal{F}_{t-1}) = \Sigma_t \succeq \sigma^2 I_n.$$

Assumption 3 formalizes the base conditional quantile curve returned by the base quantile learner as having a (possibly nonzero) conditional bias component $\mathbf{b}_t$ and a mean-zero noise component $\boldsymbol{\eta}_t$. Allowing $\mathbf{b}_t \neq 0$ is important in practice because conditional quantile learners can be biased at finite sample sizes or under model mis-specification. The covariance lower bound $\Sigma_t \succeq \sigma^2 I_n$ is a convenient non-degeneracy condition ensuring that the base conditional quantile errors contain a nontrivial amount of variability across different quantile levels, which helps to yield a clean lower bound on the amount of variance removed by the projection step.

**Assumption 4** (Truncation order growth). As $T \to \infty$, the grid size of quantile levels $n = n(T) \to \infty$, and the truncation order $K = K(n)$ in (4) satisfies: $K \to \infty$, $K/n \to 0$, and $K^{-2s} = o(\sigma^2)$.

Assumption 4 specifies an asymptotic regime where the polynomial order $K$ grows to reduce truncation bias, while remaining small relative to the grid size $n$ so that CQACP continues to smooth rather than interpolate the base conditional quantile curve. The condition $K^{-2s} = o(\sigma^2)$ is a bias–variance requirement: the squared truncation error (which decays at the order $K^{-2s}$ under Hölder($s$) smoothness) must be asymptotically negligible compared to the scale of removable estimation variability in the base conditional quantile curve. In finite samples, this motivates selecting $K$ via the cross-validation criterion described in Section 4.3.

**Assumption 5** (Conditional distribution regularity). The conditional CDF $F_t(\cdot \mid \mathcal{F}_{t-1})$ of $\widehat{\varepsilon}_t$ is continuous and strictly increasing on an interval containing $\{Q_t(p) : p \in [\underline{p}, 1 - \underline{p}]\}$ almost surely. Moreover, there exists a conditional density $f_t(\cdot \mid \mathcal{F}_{t-1})$ such that for all $p \in [\underline{p}, 1 - \underline{p}]$,

$$0 < f_{\min} \leq f_t(Q_t(p) \mid \mathcal{F}_{t-1}) \leq f_{\max} < \infty.$$

Assumption 5 is a standard identification condition for quantiles: continuity and strict monotonicity ensure conditional quantiles are well-defined and unique, while the density upper bound $f_{\max} < \infty$ guarantees that small quantile estimation error translates into small estimation error of CDF.

**Assumption 6** (High-level accuracy in the Cornish–Fisher sieve space). For the projected quantile vectors $\mathbf{q}_{t,K} := P_K \mathbf{q}_t$ and the adjusted conditional quantile estimator $\widehat{\mathbf{q}}_{t,K}^{\mathrm{adj}} = P_K \widehat{\mathbf{q}}_t$, there exists a deterministic sequence $r_n \to 0$ such that

$$\left\|\widehat{\mathbf{q}}_{t,K}^{\mathrm{adj}} - \mathbf{q}_{t,K}\right\|_n = O_{\mathbb{P}}(r_n).$$

Assumption 6 is intentionally high-level and is tailored to CQACP: it only requires vanishing error for the *sieve-projected* component of the base conditional quantile curve, i.e., the low-order Cornish–Fisher shape that CQACP ultimately uses. In contrast to a full-curve requirement such as $\|\widehat{\mathbf{q}}_t - \mathbf{q}_t\|_n = o_{\mathbb{P}}(1)$, this assumption allows the base conditional quantile learner to be noisy at fine scales across quantile levels, including quantile crossing, as long as such noise is mostly orthogonal to the sieve space and therefore removed by the projection step.

**Assumption 7** (Basis regularity for OLS). (i) (*Well-conditioning*) There exist constants $0 < c < C < \infty$ such that

$$c \;\leq\; \lambda_{\min}\Big(\frac{1}{n}Z_K^\top Z_K\Big) \;\leq\; \lambda_{\max}\Big(\frac{1}{n}Z_K^\top Z_K\Big) \;\leq\; C$$

uniformly over the values of $K$ considered;

(ii) (*Controlled basis size*) There exists a sequence $B_K$ such that

$$\sup_{z\in[-M,M]} \|\psi_K(z)\|_2 \;\leq\; B_K.$$

Assumption 7 is standard in sieve/series regression. Well-conditioning ensures that the OLS projection is numerically and statistically stable as $K$ grows, and it is typically satisfied by quasi-uniform grids over $[\underline{p}, 1 - \underline{p}]$ combined with a well-scaled polynomial basis. The bound $B_K$ controls how coefficient estimation error translates into a uniform error bound for the fitted polynomial curve on $[-M, M]$; for many normalized polynomial bases on compact intervals, one has $B_K$ growing at most polynomially in $K$.

**Assumption 8** (Sieve balance for uniform consistency). Let $s > 0$ be the Hölder smoothness exponent from the Assumption 2, and let $r_n$ and $B_K$ be as in Assumptions 6–7. Then $(1 + B_K)K^{-s} + B_K r_n \to 0$ as $K$ and $n$ diverge to infinity.

Assumption 8 is the usual approximation–estimation trade-off for series methods. The term $(1 + B_K)K^{-s}$ controls the approximation (Cornish–Fisher truncation) error under Hölder($s$) smoothness, while the term $B_K r_n$ controls the statistical error inherited from the base conditional learner after projection.

**Assumption 9** (Endpoint probabilities stay in the central region). For the target miscoverage level $\alpha \in (0, 1)$, we have that with probability tending to one,

$$\widehat{\rho}_t \alpha \in [\underline{p}, 1 - \underline{p}] \qquad \text{and} \qquad 1 - \alpha + \widehat{\rho}_t \alpha \in [\underline{p}, 1 - \underline{p}].$$

Assumption 9 ensures that CQACP evaluates the adjusted conditional quantile curve only on the central probability range where approximation and estimation are controlled. This can be enforced in practice by choosing $\underline{p}$ small and/or restricting $\widehat{\rho}_t$ to a compact subset of $(0, 1)$.

## A.2. Proof of Theorem 5.1

Throughout, fix $t$ and work conditional on $\mathcal{F}_{t-1}$. Define the OLS projector $P_K := Z_K(Z_K^\top Z_K)^{-1} Z_K^\top$, and the truncation residual (approximation error of the true quantile vector)

$$\mathbf{r}_{t,K} := (I - P_K)\mathbf{q}_t.$$

**Lemma A.1** (OLS adjustment is an orthogonal projection). *Assume $Z_K^\top Z_K$ is invertible. Then the OLS fitted values satisfy $Z_K \widehat{\theta}_{t,K} = \widehat{\mathbf{q}}_{t,K}^{\mathrm{adj}}$. Moreover, $P_K$ is an orthogonal projection onto $\mathrm{col}(Z_K)$, i.e., $P_K^\top = P_K$, $P_K^2 = P_K$, and $\mathrm{rank}(P_K) = K$.*

*Proof.* The OLS normal equations yield

$$\widehat{\theta}_{t,K} = (Z_K^\top Z_K)^{-1} Z_K^\top \widehat{\mathbf{q}}_t,$$

so $Z_K \widehat{\theta}_{t,K} = P_K \widehat{\mathbf{q}}_t = \widehat{\mathbf{q}}_{t,K}^{\mathrm{adj}}$. Symmetry and idempotence follow by direct algebra, and $\mathrm{rank}(P_K) = \mathrm{rank}(Z_K) = K$ since $Z_K$ has full column rank. $\square$

**Lemma A.2** (Risk identity for orthogonal projection). *Let $P$ be an orthogonal projection in $\mathbb{R}^n$ and define $\mathbf{r} := (I - P)\mathbf{q}$. Suppose $\widehat{\mathbf{q}} - \mathbf{q} = \mathbf{b} + \boldsymbol{\eta}$ with $\mathbb{E}[\boldsymbol{\eta} \mid \mathcal{F}] = 0$ and $\mathrm{Cov}(\boldsymbol{\eta} \mid \mathcal{F}) = \Sigma$. Then*

$$\mathbb{E}\big[\|\widehat{\mathbf{q}} - \mathbf{q}\|_n^2 - \|P\widehat{\mathbf{q}} - \mathbf{q}\|_n^2 \,\big|\, \mathcal{F}\big] = \|(I - P)\mathbf{b}\|_n^2 + \frac{1}{n}\mathrm{tr}\big((I - P)\Sigma\big) - \|\mathbf{r}\|_n^2. \tag{13}$$

*Proof.* Write $e := \widehat{\mathbf{q}} - \mathbf{q} = \mathbf{b} + \boldsymbol{\eta}$. Then

$$P\widehat{\mathbf{q}} - \mathbf{q} = P(\mathbf{q} + e) - \mathbf{q} = -(I - P)\mathbf{q} + Pe = -\mathbf{r} + Pe.$$

Since $\mathbf{r} \in \mathrm{col}(P)^{\perp}$ and $Pe \in \mathrm{col}(P)$, we have the Pythagorean identity $\|-\mathbf{r} + Pe\|_2^2 = \|\mathbf{r}\|_2^2 + \|Pe\|_2^2$, hence $\|P\widehat{\mathbf{q}} - \mathbf{q}\|_n^2 = \|\mathbf{r}\|_n^2 + \|Pe\|_n^2$. Also, by orthogonality of $P$, we have

$$\|e\|_n^2 - \|Pe\|_n^2 = \|(I - P)e\|_n^2.$$

Therefore,

$$\|\widehat{\mathbf{q}} - \mathbf{q}\|_n^2 - \|P\widehat{\mathbf{q}} - \mathbf{q}\|_n^2 = \|e\|_n^2 - \left(\|\mathbf{r}\|_n^2 + \|Pe\|_n^2\right) = \|(I - P)e\|_n^2 - \|\mathbf{r}\|_n^2.$$

Taking $\mathbb{E}[\cdot \mid \mathcal{F}]$ on both sides and using $e = \mathbf{b} + \boldsymbol{\eta}$ gives

$$\mathbb{E}[\|(I - P)e\|_n^2 \mid \mathcal{F}] = \|(I - P)\mathbf{b}\|_n^2 + \mathbb{E}[\|(I - P)\boldsymbol{\eta}\|_n^2 \mid \mathcal{F}],$$

where the cross term vanishes since $\mathbb{E}[\boldsymbol{\eta} \mid \mathcal{F}] = 0$. Finally,

$$\mathbb{E}[\|(I - P)\boldsymbol{\eta}\|_2^2 \mid \mathcal{F}] = \mathbb{E}[\boldsymbol{\eta}^{\top}(I - P)\boldsymbol{\eta} \mid \mathcal{F}] = \mathrm{tr}\big((I - P)\Sigma\big),$$

so dividing by $n$ yields (13). $\qquad\square$

**Lemma A.3** (Lower bound on variance removed by projection)**.** *If $\Sigma \succeq \sigma^2 I_n$ and $P$ is an orthogonal projection with* $\mathrm{rank}(P) = K$, *then*

$$\frac{1}{n}\mathrm{tr}\big((I - P)\Sigma\big) \geq \sigma^2\Big(1 - \frac{K}{n}\Big).$$

*Proof.* Since $\Sigma \succeq \sigma^2 I_n$ and $(I - P) \succeq 0$,

$$\mathrm{tr}\big((I - P)\Sigma\big) \geq \mathrm{tr}\big((I - P)\sigma^2 I_n\big) = \sigma^2\,\mathrm{tr}(I - P).$$

For an orthogonal projection, $\mathrm{tr}(P) = \mathrm{rank}(P) = K$, hence $\mathrm{tr}(I - P) = n - K$. Dividing by $n$ on both sides yields the result. $\qquad\square$

**Lemma A.4** (Truncation residual bound under Hölder smoothness)**.** *Under the central-grid and smoothness assumptions, there exists a constant $C > 0$ (depending only on $\underline{p}$, $s$, and the Hölder constant of $g_t$) such that*

$$\|\mathbf{r}_{t,K}\|_n^2 = \|(I - P_K)\mathbf{q}_t\|_n^2 \leq CK^{-2s}.$$

*Proof.* By the central-grid assumption, $z_i = \Phi^{-1}(p_i) \in [-M, M]$ with $M = \Phi^{-1}(1 - \underline{p})$. By Jackson's inequality for polynomial approximation bound for Hölder($s$) functions on a compact interval, there exists a polynomial $\pi_{K-1}$ of degree at most $K - 1$ such that

$$\sup_{z \in [-M,M]} |g_t(z) - \pi_{K-1}(z)| \leq C_0 K^{-s},$$

for a constant $C_0$ depending only on $(M, s)$ and the Hölder constant. Since $\psi_K$ is a polynomial basis of degree no larger than $K - 1$, $\pi_{K-1}$ can be written as $\pi_{K-1}(z) = \psi_K(z)^{\top}\theta$ for some $\theta \in \mathbb{R}^K$.

Here, define the vector $v \in \mathbb{R}^n$ with components $v_i = \pi_{K-1}(z_i)$ lies in $\mathrm{col}(Z_K)$. Since $P_K \mathbf{q}_t$ is the best $\ell_2$ approximation to $\mathbf{q}_t$ over $\mathrm{col}(Z_K)$,

$$\|(I - P_K)\mathbf{q}_t\|_2 = \min_{u \in \mathrm{col}(Z_K)} \|\mathbf{q}_t - u\|_2 \leq \|\mathbf{q}_t - v\|_2.$$

Note that $q_{t,i} = g_t(z_i)$, so

$$\|\mathbf{q}_t - v\|_2^2 = \sum_{i=1}^{n} \big(g_t(z_i) - \pi_{K-1}(z_i)\big)^2 \leq n \cdot \sup_{z \in [-M,M]} |g_t(z) - \pi_{K-1}(z)|^2 \leq nC_0^2 K^{-2s}.$$

Dividing by $n$ yields $\|(I - P_K)\mathbf{q}_t\|_n^2 \leq C_0^2 K^{-2s}$, which completes the proof with $C = C_0^2$. $\qquad\square$

*Proof of Theorem 5.1.* Apply Lemma A.2 with $P = P_K$, $\widehat{\mathbf{q}} = \widehat{\mathbf{q}}_t$, $\mathbf{q} = \mathbf{q}_t$, $\mathcal{F} = \mathcal{F}_{t-1}$, and $\mathbf{r} = \mathbf{r}_{t,K}$ to obtain

$$\mathbb{E}\Big[\|\widehat{\mathbf{q}}_t - \mathbf{q}_t\|_n^2 - \|\widehat{\mathbf{q}}_{t,K}^{\text{adj}} - \mathbf{q}_t\|_n^2 \,\Big|\, \mathcal{F}_{t-1}\Big] = \|(I - P_K)\mathbf{b}_t\|_n^2 + \frac{1}{n}\text{tr}\big((I - P_K)\Sigma_t\big) - \|\mathbf{r}_{t,K}\|_n^2.$$

Dropping the nonnegative bias term $\|(I - P_K)\mathbf{b}_t\|_n^2$ gives the lower bound

$$\mathbb{E}\Big[\|\widehat{\mathbf{q}}_t - \mathbf{q}_t\|_n^2 - \|\widehat{\mathbf{q}}_{t,K}^{\text{adj}} - \mathbf{q}_t\|_n^2 \,\Big|\, \mathcal{F}_{t-1}\Big] \geq \frac{1}{n}\text{tr}\big((I - P_K)\Sigma_t\big) - \|\mathbf{r}_{t,K}\|_n^2.$$

By the raw-error assumption and Lemma A.3,

$$\frac{1}{n}\text{tr}\big((I - P_K)\Sigma_t\big) \geq \sigma^2\Big(1 - \frac{K}{n}\Big).$$

By Lemma A.4,

$$\|\mathbf{r}_{t,K}\|_n^2 \leq CK^{-2s}.$$

Combining the two above results yields

$$\mathbb{E}\Big[\|\widehat{\mathbf{q}}_t - \mathbf{q}_t\|_n^2 - \|\widehat{\mathbf{q}}_{t,K}^{\text{adj}} - \mathbf{q}_t\|_n^2 \,\Big|\, \mathcal{F}_{t-1}\Big] \geq \sigma^2\Big(1 - \frac{K}{n}\Big) - CK^{-2s},$$

which is the first claim.

Finally, under the truncation-order growth assumption (i.e., $K/n \to 0$ and $K^{-2s} = o(\sigma^2)$), the right-hand side is strictly positive for all sufficiently large $n$, implying

$$\mathbb{E}\Big[\|\widehat{\mathbf{q}}_{t,K}^{\text{adj}} - \mathbf{q}_t\|_n^2 \,\Big|\, \mathcal{F}_{t-1}\Big] < \mathbb{E}\big[\|\widehat{\mathbf{q}}_t - \mathbf{q}_t\|_n^2 \,\big|\, \mathcal{F}_{t-1}\big].$$

This completes the proof. $\qquad\square$

### A.3. Proof of Theorem 5.2

We follow the standard long-horizon time series asymptotics used in SPCI and related work: the available history length (sample size) $T$ diverges, and all limits are taken as $T \to \infty$.

**Lemma A.5** (Uniform control of the estimation error in the sieve space). *Under Assumption 7, it holds that*

$$\sup_{p \in [\underline{p}, 1-\underline{p}]} \Big|\widehat{Q}_{t,K}^{\text{adj}}(p) - Q_{t,K}(p)\Big| \leq \frac{B_K}{\sqrt{c}}\,\big\|\widehat{\mathbf{q}}_{t,K}^{\text{adj}} - \mathbf{q}_{t,K}\big\|_n,$$

*where $c > 0$ is the lower eigenvalue bound in Assumption 7(i) and $B_K$ is the basis-size bound in Assumption 7(ii).*

*Proof.* By definition,

$$\widehat{\theta}_{t,K} - \theta_{t,K} = (Z_K^\top Z_K)^{-1} Z_K^\top(\widehat{\mathbf{q}}_t - \mathbf{q}_t).$$

Since $P_K$ is the orthogonal projector onto $\text{col}(Z_K)$, we have $Z_K^\top(I - P_K) = 0$, hence

$$Z_K^\top(\widehat{\mathbf{q}}_t - \mathbf{q}_t) = Z_K^\top P_K(\widehat{\mathbf{q}}_t - \mathbf{q}_t) = Z_K^\top(\widehat{\mathbf{q}}_{t,K}^{\text{adj}} - \mathbf{q}_{t,K}).$$

Therefore,

$$\widehat{\theta}_{t,K} - \theta_{t,K} = (Z_K^\top Z_K)^{-1} Z_K^\top(\widehat{\mathbf{q}}_{t,K}^{\text{adj}} - \mathbf{q}_{t,K}),$$

which implies

$$\|\widehat{\theta}_{t,K} - \theta_{t,K}\|_2^2 = (\widehat{\mathbf{q}}_{t,K}^{\text{adj}} - \mathbf{q}_{t,K})^\top Z_K (Z_K^\top Z_K)^{-2} Z_K^\top(\widehat{\mathbf{q}}_{t,K}^{\text{adj}} - \mathbf{q}_{t,K}) \leq \frac{1}{\lambda_{\min}(Z_K^\top Z_K)}\|\widehat{\mathbf{q}}_{t,K}^{\text{adj}} - \mathbf{q}_{t,K}\|_2^2.$$

By Assumption 7(i), $\lambda_{\min}(Z_K^\top Z_K) \geq cn$, so

$$\|\widehat{\theta}_{t,K} - \theta_{t,K}\|_2 \leq \frac{1}{\sqrt{cn}}\|\widehat{\mathbf{q}}_{t,K}^{\text{adj}} - \mathbf{q}_{t,K}\|_2 = \frac{1}{\sqrt{c}}\|\widehat{\mathbf{q}}_{t,K}^{\text{adj}} - \mathbf{q}_{t,K}\|_n.$$

For any $p \in [\underline{p}, 1 - \underline{p}]$,

$$\left|\widehat{Q}_{t,K}^{\mathrm{adj}}(p) - Q_{t,K}(p)\right| = \left|\psi_K(z_p)^\top(\widehat{\theta}_{t,K} - \theta_{t,K})\right| \leq \|\psi_K(z_p)\|_2 \|\widehat{\theta}_{t,K} - \theta_{t,K}\|_2 \leq \frac{B_K}{\sqrt{c}} \|\widehat{\mathbf{q}}_{t,K}^{\mathrm{adj}} - \mathbf{q}_{t,K}\|_n,$$

where we have used Assumption 7(ii). Taking the supremum over $p \in [\underline{p}, 1 - \underline{p}]$ yields the claim. $\qquad\square$

**Lemma A.6** (Sieve approximation error). *Suppose Assumptions 1, 2 and 7 hold. Then there exists a constant $C > 0$ such that*

$$\sup_{u \in [p,\, 1-p]} \left|Q_{t,K}(u) - Q_t(u)\right| \leq C\,(1 + B_K)\,K^{-s}.$$

*In particular,* $\sup_{u \in [p,\, 1-p]} |Q_{t,K}(u) - Q_t(u)| = O((1 + B_K)K^{-s})$.

*Proof.* By Assumption 2, $g_t$ belongs to a Hölder($s$) class on $[-M, M]$. By Jackson's inequality, there exists a polynomial $\pi_{K-1}$ of degree at most $K - 1$ such that

$$\sup_{z \in [-M, M]} \left|g_t(z) - \pi_{K-1}(z)\right| \leq C_0 K^{-s}, \tag{14}$$

for a constant $C_0$ depending only on $M$ and $s$.

Since $\pi_{K-1}$ has degree at most $K - 1$, it lies in the span of the basis $\psi_K$, so there exists $\theta_\pi \in \mathbb{R}^K$ such that $\pi_{K-1}(z) = \psi_K(z)^\top \theta_\pi$. Let $z_i := \Phi^{-1}(p_i)$ and define the vector $\pi \in \mathbb{R}^n$ by $\pi_i := \pi_{K-1}(z_i)$, so $\pi = Z_K \theta_\pi \in \mathrm{col}(Z_K)$.

Recall that $\theta_{t,K}$ is defined by the least-squares projection on the grid:

$$\theta_{t,K} \in \arg\min_{\theta \in \mathbb{R}^K} \|q_t - Z_K\theta\|_2^2, \qquad \text{equivalently} \qquad Z_K\theta_{t,K} = P_K q_t,$$

where $P_K$ is the orthogonal projector onto $\mathrm{col}(Z_K)$. Hence the residual $r = q_t - Z_K\theta_{t,K}$ is orthogonal to $\mathrm{col}(Z_K)$, and in particular $r^\top(\pi - Z_K\theta_{t,K}) = 0$. Therefore, by the Pythagorean identity,

$$\|q_t - \pi\|_2^2 = \|q_t - Z_K\theta_{t,K}\|_2^2 + \|\pi - Z_K\theta_{t,K}\|_2^2 \geq \|\pi - Z_K\theta_{t,K}\|_2^2,$$

so

$$\|Z_K(\theta_{t,K} - \theta_\pi)\|_2 = \|Z_K\theta_{t,K} - \pi\|_2 \leq \|q_t - \pi\|_2. \tag{15}$$

Note that $q_{t,i} = Q_t(p_i) = g_t(z_i)$, so by (14),

$$\|q_t - \pi\|_2^2 = \sum_{i=1}^n \big(g_t(z_i) - \pi_{K-1}(z_i)\big)^2 \leq n\Big(\sup_{z \in [-M,M]} |g_t(z) - \pi_{K-1}(z)|\Big)^2 \leq n C_0^2 K^{-2s}.$$

Combining with (15) yields

$$\|Z_K(\theta_{t,K} - \theta_\pi)\|_2 \leq \sqrt{n}\, C_0 K^{-s}.$$

By Assumption 7(i), $\lambda_{\min}\big(\frac{1}{n} Z_K^\top Z_K\big) \geq c$, hence $\lambda_{\min}(Z_K^\top Z_K) \geq cn$, and thus

$$\|\theta_{t,K} - \theta_\pi\|_2 \leq \frac{1}{\sqrt{\lambda_{\min}(Z_K^\top Z_K)}} \|Z_K(\theta_{t,K} - \theta_\pi)\|_2 \leq \frac{1}{\sqrt{cn}} \cdot \sqrt{n}\, C_0 K^{-s} = \frac{C_0}{\sqrt{c}}\, K^{-s}.$$

Now fix any $u \in [\underline{p}, 1 - \underline{p}]$ and write $z_u := \Phi^{-1}(u) \in [-M, M]$. Using $Q_{t,K}(u) = \psi_K(z_u)^\top \theta_{t,K}$ and $\pi_{K-1}(z_u) = \psi_K(z_u)^\top \theta_\pi$,

$$|Q_{t,K}(u) - Q_t(u)| = |\psi_K(z_u)^\top \theta_{t,K} - g_t(z_u)| \leq |\psi_K(z_u)^\top(\theta_{t,K} - \theta_\pi)| + |\pi_{K-1}(z_u) - g_t(z_u)|.$$

The second term is bounded by $C_0 K^{-s}$ from (14). For the first term, Cauchy–Schwarz and Assumption 7(ii) give

$$|\psi_K(z_u)^\top(\theta_{t,K} - \theta_\pi)| \leq \|\psi_K(z_u)\|_2 \|\theta_{t,K} - \theta_\pi\|_2 \leq B_K \cdot \frac{C_0}{\sqrt{c}}\, K^{-s}.$$

Taking the supremum over $u \in [\underline{p}, 1 - \underline{p}]$ yields

$$\sup_{u \in [\underline{p}, 1-\underline{p}]} |Q_{t,K}(u) - Q_t(u)| \leq C_0 K^{-s} + \frac{C_0}{\sqrt{c}} B_K K^{-s} \leq C(1 + B_K) K^{-s},$$

for a constant $C$ depending only on $C_0$ and $c$. This completes the proof. $\qquad\square$

**Lemma A.7** (Uniform consistency of the CQACP adjusted conditional quantile curve). *Suppose that Assumptions 1, 2, 6, 7, and 8 hold. Then we have*

$$\sup_{p \in [\underline{p}, \, 1-\underline{p}]} \left| \widehat{Q}_{t,K}^{\mathrm{adj}}(p) - Q_t(p) \right| \xrightarrow{\mathbb{P}} 0 \qquad as \ T \to \infty.$$

*Proof.* By the triangle inequality,

$$\sup_{p \in [\underline{p}, \, 1-\underline{p}]} \left| \widehat{Q}_{t,K}^{\mathrm{adj}}(p) - Q_t(p) \right| \leq \sup_{p \in [\underline{p}, \, 1-\underline{p}]} \left| \widehat{Q}_{t,K}^{\mathrm{adj}}(p) - Q_{t,K}(p) \right| + \sup_{p \in [\underline{p}, \, 1-\underline{p}]} \left| Q_{t,K}(p) - Q_t(p) \right|.$$

By Lemma A.5,

$$\sup_{p \in [\underline{p}, \, 1-\underline{p}]} \left| \widehat{Q}_{t,K}^{\mathrm{adj}}(p) - Q_{t,K}(p) \right| \leq \frac{B_K}{\sqrt{c}} \left\| \widehat{\mathbf{q}}_{t,K}^{\mathrm{adj}} - \mathbf{q}_{t,K} \right\|_n.$$

By Assumption 6, $\left\| \widehat{\mathbf{q}}_{t,K}^{\mathrm{adj}} - \mathbf{q}_{t,K} \right\|_n = O_{\mathbb{P}}(r_n)$, hence

$$\sup_{p \in [\underline{p}, \, 1-\underline{p}]} \left| \widehat{Q}_{t,K}^{\mathrm{adj}}(p) - Q_{t,K}(p) \right| = O_{\mathbb{P}}(B_K r_n),$$

absorbing the constant $1/\sqrt{c}$.

By Lemma A.6, there exists a constant $C > 0$ such that

$$\sup_{p \in [\underline{p}, \, 1-\underline{p}]} \left| Q_{t,K}(p) - Q_t(p) \right| \leq C(1 + B_K) K^{-s} = O\big((1 + B_K) K^{-s}\big).$$

Combining the results above yields

$$\sup_{p \in [\underline{p}, \, 1-\underline{p}]} \left| \widehat{Q}_{t,K}^{\mathrm{adj}}(p) - Q_t(p) \right| = O_{\mathbb{P}}(B_K r_n) + O\big((1 + B_K) K^{-s}\big).$$

Hence, the claim follows by Assumption 8. $\qquad\square$

**Lemma A.8** (Quantile error implies CDF calibration error). *Suppose that Assumption 5 holds. Then, for any $p \in [\underline{p}, 1 - \underline{p}]$, we have*

$$\left| F_t(\widehat{Q}_{t,K}^{\mathrm{adj}}(p) \mid \mathcal{F}_{t-1}) - p \right| \leq f_{\max} \left| \widehat{Q}_{t,K}^{\mathrm{adj}}(p) - Q_t(p) \right|.$$

*Consequently,*

$$\sup_{p \in [\underline{p}, 1-\underline{p}]} \left| F_t(\widehat{Q}_{t,K}^{\mathrm{adj}}(p) \mid \mathcal{F}_{t-1}) - p \right| \leq f_{\max} \sup_{p \in [\underline{p}, 1-\underline{p}]} \left| \widehat{Q}_{t,K}^{\mathrm{adj}}(p) - Q_t(p) \right|.$$

*Proof.* For any $p \in [\underline{p}, 1 - \underline{p}]$, by definition of the conditional quantile and Assumption 5, we have

$$F_t(Q_t(p) \mid \mathcal{F}_{t-1}) = p.$$

Let $\Delta(p) := \widehat{Q}_{t,K}^{\mathrm{adj}}(p) - Q_t(p)$. By the mean value theorem applied to the conditional CDF in its score argument, there exists a (random) point $\xi(p)$ between $Q_t(p)$ and $Q_t(p) + \Delta(p)$ such that

$$F_t(Q_t(p) + \Delta(p) \mid \mathcal{F}_{t-1}) - F_t(Q_t(p) \mid \mathcal{F}_{t-1}) = f_t(\xi(p) \mid \mathcal{F}_{t-1}) \Delta(p).$$

Using $F_t(Q_t(p) \mid \mathcal{F}_{t-1}) = p$ and the upper density bound $f_t(\xi(p) \mid \mathcal{F}_{t-1}) \leq f_{\max}$ in Assumption 5 yields

$$\left| F_t(\widehat{Q}_{t,K}^{\mathrm{adj}}(p) \mid \mathcal{F}_{t-1}) - p \right| = |f_t(\xi(p) \mid \mathcal{F}_{t-1})| |\Delta(p)| \leq f_{\max} |\Delta(p)|.$$

Taking the supremum over $p \in [\underline{p}, 1 - \underline{p}]$ gives the second inequality. $\qquad\square$

*Proof of Theorem 5.2.* Define the endpoint probabilities as

$$p_L := \widehat{\rho}_t \alpha, \qquad p_U := 1 - \alpha + \widehat{\rho}_t \alpha,$$

so that $p_U - p_L = 1 - \alpha$. Let $\mathcal{E}_t$ denote the event in Assumption 9 that $p_L, p_U \in [\underline{p}, 1 - \underline{p}]$. By Assumption 9, $\mathbb{P}(\mathcal{E}_t) \to 1$ as $T \to \infty$.

On $\mathcal{E}_t$, the CQACP interval endpoints are evaluated in the central region. Moreover, since $\widehat{Q}_{t,K}^{\mathrm{adj}}(\cdot)$ is a quantile curve (and can be monotone-enforced if needed), the interval is well-defined, and conditional coverage can be written in terms of the conditional CDF:

$$\mathbb{P}\big(Y_t \in \widehat{C}_{\alpha,t-1}(X_t) \mid \mathcal{F}_{t-1}\big) = \mathbb{P}\Big(\widehat{\varepsilon}_t \in \big[\widehat{Q}_{t,K}^{\mathrm{adj}}(p_L), \widehat{Q}_{t,K}^{\mathrm{adj}}(p_U)\big] \,\Big|\, \mathcal{F}_{t-1}\Big)$$
$$= F_t\big(\widehat{Q}_{t,K}^{\mathrm{adj}}(p_U) \mid \mathcal{F}_{t-1}\big) - F_t\big(\widehat{Q}_{t,K}^{\mathrm{adj}}(p_L) \mid \mathcal{F}_{t-1}\big).$$

Therefore, on $\mathcal{E}_t$,

$$\left| \mathbb{P}\big(Y_t \in \widehat{C}_{\alpha,t-1}(X_t) \mid \mathcal{F}_{t-1}\big) - (1-\alpha) \right|$$
$$= \left| \big[ F_t(\widehat{Q}_{t,K}^{\mathrm{adj}}(p_U) \mid \mathcal{F}_{t-1}) - p_U \big] - \big[ F_t(\widehat{Q}_{t,K}^{\mathrm{adj}}(p_L) \mid \mathcal{F}_{t-1}) - p_L \big] \right|$$
$$\leq \left| F_t(\widehat{Q}_{t,K}^{\mathrm{adj}}(p_U) \mid \mathcal{F}_{t-1}) - p_U \right| + \left| F_t(\widehat{Q}_{t,K}^{\mathrm{adj}}(p_L) \mid \mathcal{F}_{t-1}) - p_L \right|.$$

Applying Lemma A.8 (with the $\mathcal{F}_{t-1}$-measurable random probabilities $p_U, p_L$) yields

$$\left| F_t(\widehat{Q}_{t,K}^{\mathrm{adj}}(p_U) \mid \mathcal{F}_{t-1}) - p_U \right| + \left| F_t(\widehat{Q}_{t,K}^{\mathrm{adj}}(p_L) \mid \mathcal{F}_{t-1}) - p_L \right| \leq 2 f_{\max} \sup_{p \in [\underline{p}, 1-\underline{p}]} \left| \widehat{Q}_{t,K}^{\mathrm{adj}}(p) - Q_t(p) \right|.$$

By Lemma A.7, the right-hand side converges to 0 in probability as $T \to \infty$. Since $\mathbb{P}(\mathcal{E}_t) \to 1$, the same conclusion holds without conditioning on $\mathcal{E}_t$:

$$\left| \mathbb{P}\big(Y_t \in \widehat{C}_{\alpha,t-1}(X_t) \mid \mathcal{F}_{t-1}\big) - (1-\alpha) \right| \xrightarrow{\mathbb{P}} 0.$$

This completes the proof. $\square$

# B. Additional Ablation Discussion

**Relation to monotone rearrangement.** Monotone rearrangement is a principled ex-post correction for enforcing non-crossing quantile curves after separate quantile estimators have been obtained (Chernozhukov et al., 2010). However, it does not share information across quantile levels or reduce the noise in the raw estimators. Our goal is different. To be specific, the CF projection first pools information across the entire quantile grid through a low-dimensional moment-parameterized representation, thereby denoising unstable raw quantile estimators, especially in the tails. The monotonicity constraint then enforces coherence of the adjusted curve, and the shared asymmetry ratio ensures interval nesting across significance levels. Thus, in CQACP, cross-quantile information sharing is the main stability mechanism, while monotonicity and shared asymmetry provide non-crossing and nestedness.

## B.1. Ablation: Monotone rearrangement alone

We use SPCI-MR to denote SPCI followed by the monotone rearrangement of the estimated quantile curve. Table B.4 reports an ablation comparing SPCI, SPCI with monotone rearrangement, and CQACP. This ablation separates the effect of the ex-post monotonicity correction from the cross-quantile CF projection used by CQACP. The comparison is conducted on the Realized Volatility dataset. As demonstrated in Table B.4, SPCI-MR improves modestly over SPCI, but CQACP is closer to nominal coverage and has lower variability, suggesting that cross-quantile information sharing, not monotonicity correction alone, is the main source of gain.

## B.2. Sensitivity to the CF truncation order

The truncation order $K$ controls the bias–variance tradeoff of the CF approximation: Smaller $K$ gives a smoother but more biased quantile curve, while larger $K$ is more flexible but can be noisier. The main method uses cross-validation over $K \in \{3, \ldots, 10\}$. We report a sensitivity check on the Realized Volatility dataset.

*Table B.4.* Ablation comparing monotone rearrangement and CQACP on the Realized Volatility dataset.

| Method | Coverage | Std. | Width | Std. |
|--------|----------|------|-------|------|
| SPCI | 87.4% | 1.3% | 0.8% | 0.0% |
| SPCI-MR | 88.2% | 1.1% | 0.8% | 0.0% |
| CQACP | 89.9% | 0.4% | 0.8% | 0.0% |

*Table B.5.* Sensitivity of CQACP to the CF truncation order $K$ on the Realized Volatility dataset.

| $K$ | Coverage | Std. | Width | Std. |
|-----|----------|------|-------|------|
| 3 | 88.7% | 0.3% | 0.8% | 0.0% |
| 4 | 90.1% | 0.2% | 0.9% | 0.0% |
| 5 | 90.1% | 0.4% | 0.9% | 0.0% |
| 6 | 90.4% | 0.6% | 0.9% | 0.0% |

From Table B.5, we find that the performance of CQACP is stable for $K = 4, 5, 6$, while too small $K$ may underfit the quantile curve.

## C. Synthetic Stress Tests under Heavy and Changing Tails

We further stress-test the CF approximation under heavy-tailed and changing-tail behavior. The synthetic data are generated from an AR-type conditional mean with GARCH(1,1) volatility. The heavy-tail design uses Student-$t$ innovations with 5 degrees of freedom, rescaled when necessary to keep the innovation variance comparable across designs. The changing-tail design alternates the innovation degrees of freedom between 20 and 5 every 250 time points. Each Monte Carlo sample has length 2500 and is split chronologically into 80% training and 20% testing; results are averaged over 100 repetitions. We use the same implementation as in the main experiments.

*Table C.6.* Synthetic stress tests at $\alpha = 10\%$. Standard deviations over 100 Monte Carlo repetitions are reported in adjacent Std. columns.

| Method | Heavy Tail | | | | Changing Tail | | | |
|--------|----------|------|-------|------|----------|------|-------|------|
| | Coverage | Std. | Width | Std. | Coverage | Std. | Width | Std. |
| EnbPI | 85.0% | 3.4% | 2.72 | 0.25 | 84.9% | 2.7% | 2.82 | 0.25 |
| SPCI | 88.7% | 2.4% | 2.71 | 0.30 | 87.9% | 2.2% | 2.78 | 0.30 |
| CQACP | 89.7% | 2.3% | 2.76 | 0.30 | 89.6% | 2.1% | 2.84 | 0.30 |

*Table C.7.* Synthetic stress tests at $\alpha = 5\%$. Standard deviations over 100 Monte Carlo repetitions are reported in adjacent Std. columns.

| Method | Heavy Tail | | | | Changing Tail | | | |
|--------|----------|------|-------|------|----------|------|-------|------|
| | Coverage | Std. | Width | Std. | Coverage | Std. | Width | Std. |
| EnbPI | 90.6% | 2.5% | 3.50 | 0.34 | 90.2% | 2.3% | 3.53 | 0.38 |
| SPCI | 92.4% | 1.8% | 3.36 | 0.39 | 92.8% | 1.8% | 3.39 | 0.41 |
| CQACP | 94.6% | 1.6% | 3.39 | 0.43 | 94.7% | 1.5% | 3.39 | 0.42 |

From Tables C.6–C.7, we find that CQACP achieves coverage closest to target with comparable widths, suggesting CF adjustment remains effective for conformal tail quantiles under these challenging tail designs.

### C.1. Oracle quantile MSE by quantile region

To isolate quantile-estimation error, we repeat the synthetic stress tests using the oracle conditional mean as the point predictor and compare the raw SPCI quantile curve with CQACP. The quantile grid is $p \in \{0.01, 0.02, \ldots, 0.99\}$. We define

the relative MSE reduction as

$$\Delta\text{MSE} = 1 - \frac{\text{MSE}_{\text{CQACP}}}{\text{MSE}_{\text{SPCI}}}.$$

In Table C.8, we report results on the full grid, the center region $p \in [0.40, 0.60]$, and the tail region $p \in [0.01, 0.10] \cup [0.90, 0.99]$. As observed from this table, the reduction is much larger in the tail region than in the center region, supporting the claim that CQACP denoises most strongly where raw conditional quantile estimators are unstable.

*Table C.8.* Oracle quantile MSE by quantile region. Standard deviations over Monte Carlo repetitions are reported in adjacent Std. columns.

| Region | Design | SPCI MSE | Std. | CQACP MSE | Std. | $\Delta$MSE | Std. |
|---|---|---|---|---|---|---|---|
| Full grid | Heavy-tail | 0.086 | 0.061 | 0.073 | 0.053 | 14.99% | 2.36% |
| Full grid | Changing-tail | 0.089 | 0.095 | 0.075 | 0.077 | 14.31% | 2.01% |
| Center | Heavy-tail | 0.016 | 0.007 | 0.015 | 0.006 | 2.71% | 1.88% |
| Center | Changing-tail | 0.021 | 0.012 | 0.019 | 0.009 | 3.01% | 1.06% |
| Tail | Heavy-tail | 0.282 | 0.214 | 0.227 | 0.184 | 19.34% | 3.93% |
| Tail | Changing-tail | 0.253 | 0.235 | 0.204 | 0.205 | 18.26% | 3.45% |

## D. Additional Experiment Details

### D.1. Data description

The *Realized Volatility* dataset contains daily realized volatility of the S&P 500 index from January 1, 2010 to December 31, 2024. As realized volatility is computed as the standard deviation of high-frequency intraday returns (e.g., 1-minute returns), it is always used to measure the risk of financial asset. Following the recommended settings in Corsi (2009), the label for this dataset is the one-day-ahead realized volatility, and the features consist of daily, weekly (5-day), and monthly (21-day) moving averages of realized volatility.

While the first dataset focuses on risk dynamics, the second and third datasets examine return dynamics. The second dataset, *Index Return*, contains daily S&P 500 index returns over the same period, while the third dataset, *Exchange Rates*, consists of daily USD–CNY exchange rate returns from January 1, 2010 to March 1, 2025. For both datasets, features are constructed using daily, weekly, and monthly moving averages, and one-day-ahead returns are used as labels. Compared with index returns, the exchange rate series exhibits more stable dynamics, allowing us to evaluate the performance of CQACP in a relatively stationary setting.

In addition to financial time series, we consider the electricity market dataset, as in Xu & Xie (2023). The *Electricity* dataset records electricity usage and pricing in New South Wales and Victoria, Australia, at a 30-minute frequency over a 2.5-year period from 1996 to 1999. In this dataset, we predict the quantity of electricity transferred between the two states, illustrating the applicability of CQACP beyond financial domains.

The three financial datasets are split into training, validation, and test sets in chronological order, with proportions of 60%, 20%, and 20%, respectively. By contrast, the electricity dataset follows the same approach in Xu & Xie (2023) to split training, validation and testing. For non-machine-learning methods, the validation set is merged into the training set.

### D.2. Implementation details

For each existing conformal prediction method, we adopt the point prediction model $\widehat{f}(\cdot)$ recommended in the original reference. For instance, SPCI uses a random forest for point prediction. To ensure a fair comparison with SPCI, CQACP also employs a random forest as the point predictor.

Meanwhile, only SPCI and CQACP require an additional quantile learner. For the results reported in Table 1 and Table 2, we use quantile random forest (Shiraishi et al., 2024) as the quantile learner for both methods. For Table 3, we consider different quantile learners to examine the impact of the quantile model choice.

To select the truncation order $K$ for CQACP, we minimize the cross-validation criterion in (12) over the candidate set $\mathcal{K} = \{3, 4, 5, \ldots, 10\}$ in this paper. We also evaluate each method with 5 independent runs to account for possible stochasticity.

All other unreported implementation details follow the settings recommended in the original references.

## E. Additional Experiment: Impact of Significance Level

We have reported the impact of the significance level on CQACP, EnbPI, and SPCI in Table 2. In this section, we present results for the remaining baselines in Table E.1. Overall, ACI-type methods that adaptively adjust the target coverage level deliver empirical coverage close to the nominal level across different choices of $\alpha$. However, they do so at a substantial cost in efficiency: their prediction intervals are consistently much wider, typically more than twice the width of those produced by CQACP (and ACI even yields unbounded widths). In contrast, the probabilistic forecasting models (DeepAR and TFT) are less reliable, with empirical coverage that can deviate from the nominal level and intervals that are often wider than those of CQACP, leading to inferior performance in both coverage and width.

*Table E.1.* Empirical coverage and average width of prediction intervals across different choices of $\alpha$. Ideally, the coverage should be close to the nominal confidence level $1 - \alpha$, while the average width should be as small as possible. The standard deviation of each method over five independent trials is reported in parentheses, where "-" indicates zero standard deviation.

| Method | Realized Volatility | | | | Index Return | | | | Exchange Rates | | | | Electricity | | | |
|---|---|---|---|---|---|---|---|---|---|---|---|---|---|---|---|---|
| | Coverage | | Width | | Coverage | | Width | | Coverage | | Width | | Coverage | | Width | |
| | | | | | | | Panel A: $\alpha = 10\%$ | | | | | | | | | |
| ACI | 89.9% | - | $\infty$ | - | 90.1% | - | $\infty$ | - | 90.0% | - | $\infty$ | - | 90.0% | - | $\infty$ | - |
| OGD | 89.9% | - | 44.9% | - | 89.9% | - | 44.9% | - | 90.0% | - | 45.0% | - | 90.0% | - | 48.3% | - |
| SF-OGD | 89.9% | - | 4.3% | - | 89.9% | - | 5.9% | - | 90.0% | - | 3.2% | - | 90.0% | - | 16.2% | - |
| PID | 90.1% | - | 2.6% | - | 90.1% | - | 7.0% | - | 90.2% | - | 1.4% | - | 90.0% | - | 25.0% | - |
| ECI | 89.9% | - | 2.3% | - | 89.9% | - | 6.6% | - | 90.0% | - | 1.3% | - | 90.0% | - | 24.1% | - |
| DeepAR | 89.9% | (2.9%) | 0.9% | (0.1%) | 85.3% | (1.0%) | 3.1% | (0.1%) | 89.3% | (0.6%) | 0.9% | (0.0%) | 93.0% | (1.0%) | 16.7% | (1.0%) |
| TFT | 92.3% | (5.7%) | 1.1% | (0.2%) | 88.4% | (1.0%) | 3.3% | (0.1%) | 87.9% | (0.5%) | 0.8% | (0.0%) | 91.6% | (1.1%) | 21.3% | (1.3%) |
| | | | | | | | Panel B: $\alpha = 5\%$ | | | | | | | | | |
| ACI | 95.0% | - | $\infty$ | - | 95.0% | - | $\infty$ | - | 95.0% | - | $\infty$ | - | 95.0% | - | $\infty$ | - |
| OGD | 95.0% | - | 47.6% | - | 95.0% | - | 47.7% | - | 95.0% | - | 47.4% | - | 95.0% | - | 54.5% | - |
| SF-OGD | 95.0% | - | 7.0% | - | 95.0% | - | 8.8% | - | 95.0% | - | 5.2% | - | 95.0% | - | 20.9% | - |
| PID | 95.1% | - | 3.0% | - | 95.0% | - | 8.1% | - | 95.2% | - | 1.8% | - | 95.0% | - | 29.5% | - |
| ECI | 95.0% | - | 2.7% | - | 95.1% | - | 7.7% | - | 95.0% | - | 1.5% | - | 95.0% | - | 28.4% | - |
| DeepAR | 92.5% | (2.7%) | 1.0% | (0.2%) | 91.0% | (0.7%) | 3.7% | (0.1%) | 92.7% | (0.7%) | 1.1% | (0.0%) | 95.5% | (0.7%) | 19.9% | (1.1%) |
| TFT | 98.0% | (1.4%) | 1.5% | (0.2%) | 96.1% | (0.7%) | 4.8% | (0.1%) | 94.3% | (0.4%) | 1.1% | (0.0%) | 96.8% | (1.2%) | 29.0% | (2.0%) |
| | | | | | | | Panel C: $\alpha = 2.5\%$ | | | | | | | | | |
| ACI | 97.5% | - | $\infty$ | - | 97.5% | - | $\infty$ | - | 97.5% | - | $\infty$ | - | 97.5% | - | $\infty$ | - |
| OGD | 97.5% | - | 48.7% | - | 97.5% | - | 49.6% | - | 97.5% | - | 48.8% | - | 97.5% | - | 59.5% | - |
| SF-OGD | 97.5% | - | 10.6% | - | 97.5% | - | 12.2% | - | 97.5% | - | 8.0% | - | 97.5% | - | 25.5% | - |
| PID | 97.6% | - | 3.5% | - | 97.5% | - | 8.6% | - | 97.7% | - | 1.9% | - | 97.5% | - | 34.8% | - |
| ECI | 97.5% | - | 3.0% | - | 97.5% | - | 8.3% | - | 97.6% | - | 1.8% | - | 97.5% | - | 32.7% | - |
| DeepAR | 94.2% | (2.2%) | 1.2% | (0.2%) | 94.1% | (1.0%) | 4.2% | (0.1%) | 95.2% | (0.3%) | 1.2% | (0.0%) | 96.9% | (0.6%) | 22.8% | (1.3%) |
| TFT | 99.2% | (0.5%) | 1.9% | (0.1%) | 98.6% | (0.8%) | 6.2% | (0.7%) | 97.4% | (0.1%) | 1.4% | (0.0%) | 98.1% | (1.7%) | 34.6% | (4.2%) |

