# OpenReview forum: "Conditional Quantile Adjusted Conformal Prediction for Time Series"
_ICML.cc/2026/Conference — ICML 2026 regular_

### Official Review · Reviewer_DiDT · 2026-02-27

**Soundness:** 3
**Presentation:** 2
**Significance:** 2
**Originality:** 3
**Overall Recommendation:** 4
**Confidence:** 3

**Summary:**

The authors address the problem of quantile crossing in the time series setup where the data are not exchangeable. Stability over different confidence levels is obtained by fitting a monotonic regression model at each time step.

**Compliance With Llm Reviewing Policy:**

Affirmed.

**Final Justification:**

I thank the authors for the rebuttal and the further clarifications, which addressed all my concerns. I raised my score to 4.

**Key Questions For Authors:**

- What do you mean by *under serial dependence*?
- Can the approach provide finite-sample guarantees, analogous to CP?
- Does smoothness over different confidence levels imply smoothness over time?
- Are there regularity assumptions on the data-generating distribution?
- Should the nested-interval property be imposed *by hand*? Doesn’t this follow from the monotonicity of the conformity score function?
- Why is using weighted empirical distributions unstable? Are all QRF quantile estimators based on reweighting techniques?
- According to Section 5, the method’s improvements are *in terms of mean squared error (MSE)*. Is the mean over different confidence levels? If so, how does this relate to the claim that the proposed approach performs better than SPCI on the estimation of tail conditional quantiles?
- How can the methods have *zero standard deviation* across different runs? Was any randomness expected?
- Is it fair to compare methods that provide finite-sample guarantees with methods that do not?

**Limitations:**

Yes

**Strengths And Weaknesses:**

**Strengths**
- Conformal Prediction in the non-exchangeable and time-varying setup is an urgent and challenging problem.
- The use of monotonic regression as a regularization scheme is interesting.
- The proposed algorithm performs well in the experiments.

**Weaknesses**
- The work may be justified better. Avoiding quantile crossing when considering different significance levels does not seem to be a strong motivation.
- It is unclear whether the method imposes or achieves smoothness in the time direction (at a fixed significance level).
- The method is called *conformal* in the title, but coverage is only guaranteed asymptotically.
- It is OK to move technical proofs to the appendix. It is more questionable to move the method’s assumptions there.

---

> ### Author Rebuttal · Authors · 2026-03-29
>
> We thank you very much for the thorough and detailed comments. Our point-by-point responses are below.
>
> ***Q1: What do you mean by under serial dependence?***
>
> "Serial dependence" means the nonconformity scores are not independent across time: The conditional distribution of $\widehat{\epsilon}_t$ depends on past values. This arises because (i) the process $Y_t$ itself has temporal dependence and (ii) residuals inherit dependence from time-varying higher-order moments. Our theoretical results (Theorems 1–2) allow this structure rather than requiring i.i.d. or exchangeability. We will define this explicitly in the revision.
>
> ***Q2: Can the approach provide finite-sample guarantees, analogous to CP?***
>
> Exact finite-sample conditional coverage is generally not available for dependent data without very strong assumptions. Classical conformal guarantees rely on exchangeability. Under dependence, one can instead characterize the coverage gap or target long-run average coverage, but these are weaker than the conditional coverage notion we study. Developing non-asymptotic conditional guarantees under serial dependence remains an important open problem.
>
> ***Q3: Does smoothness over different confidence levels imply smoothness over time?***
>
> No. These two concepts are distinct notions. Smoothness over confidence levels concerns the shape of the quantile curve  $Q_{t}(p)$ as a function of quantile level $p$ at a fixed time $t$, while the smoothness over time concerns the evolution of $Q_{t}(p)$ over time $t$ at a fixed quantile level $p$.
>
> ***Q4: Are there regularity assumptions on the data-generating distribution?***
>
> Yes. The key Assumptions 2, 3 and 5 are stated in Appendix A.1. All are standard in quantile regression and sieve estimation. In the revision, we will move a concise summary of the main assumptions into the main text and keep the technical details in the appendix.
>
> ***Q5: Should the nested-interval property be imposed by hand? Doesn't this follow from the monotonicity of the conformity score function?***
>
> Nestedness does not automatically follow from score monotonicity in the sequential setting. In SPCI, the prediction interval is two-sided with a level-specific asymmetry parameter, and quantile estimates at different levels are obtained independently. Both factors can produce non-nested intervals. CQACP resolves this by (i) fitting a monotone quantile curve and (ii) using a single shared $\hat{ρ}_t$ across all levels (Eq.13). Both features are necessary for nestedness.
>
> ***Q6: Why is using weighted empirical distributions unstable? Are all QRF quantile estimators based on reweighting techniques?***
>
> QRF estimates conditional quantiles via a weighted empirical distribution, where weights come from the forest's leaf structure. Tail instability arises because (1) the effective local sample size is limited by leaf sizes, so extreme quantiles (e.g., p = 0.01) rely on very few observations, and (2) the resulting quantile function is piecewise constant with abrupt jumps.
>
> ***Q7: According to Section 5, the method's improvements are in terms of MSE. Is the mean over different confidence levels? If so, how does this relate to the claim that the proposed approach performs better than SPCI on the estimation of tail conditional quantiles?***
>
> The MSE in Theorem 1 is the average squared error across all quantile levels in the grid. The tails are the noisiest part of the base estimates and contribute the most to the removable variance $σ^2(1 − K/n)$, so they benefit disproportionately from the denoising projection. Empirically, at $α$ = 2.5% (Table 2, Panel C), CQACP achieves 97.4% coverage vs. SPCI's 95.8% on Electricity (target: 97.5%), confirming that the advantage is most pronounced where tail estimation is hardest.
>
> ***Q8: How can the methods have zero standard deviation across different runs? Was any randomness expected?***
>
> Methods marked "−" (ACI-type methods) are fully deterministic: They adjust the coverage level using deterministic update rules. Given the same data and initialization, they produce identical intervals. EnbPI, SPCI, and CQACP use random forests, producing non-zero std. DeepAR and TFT have non-zero std from stochastic optimization.
>
> ***Q9: Is it fair to compare methods that provide finite-sample guarantees with methods that do not?***
>
> The comparison is fair empirically. All methods are evaluated on the same dataset with the same metric. The comparison reveals practical trade-offs: ACI-type methods target at long-run marginal coverage by being conservative, sometimes producing unbounded widths (Table 1). SPCI, our most direct competitor, also provides only asymptotic guarantees, making the comparison natural. Cross-guarantee-type comparisons are standard in the literature (e.g., Xu et al., 2023; Wu et al., 2025).
>
> **Due to space limits, we can address the rest during discussion. We hope our responses and planned revisions have successfully addressed your concerns.**

---

> > ### Author Rebuttal · Reviewer_DiDT · 2026-04-02
> >
> > - Why is the method called *conformal*?
> >
> > - Would it be possible to provide it with ACI-like, *weaker*, guarantees?
> >
> > - Could the statement that tails *benefit disproportionately from the denoising projection* be supported numerically by reporting the MSE over $1 - \alpha > 0.1$ or so?

---

> > > ### Author Response · Authors · 2026-04-03
> > >
> > > Thank you for your additional questions, which will help us improve the paper. Our point-by-point responses are below.
> > >
> > > ***Q1: Why is the method called conformal?***
> > >
> > > ***A1:*** We call the method **conformal** because it retains the conformal prediction paradigm: given a black-box point predictor $\widehat{f}$, we define a nonconformity score, here the prediction residual $\widehat{\epsilon}_t = Y_t - \widehat{f}(X_t)$, estimate quantiles of the score distribution, and construct the prediction interval by calibrating these score quantiles around the point forecast.
> > >
> > > In our paper, **CQACP keeps exactly this score-based sequential calibration structure**; its role is not to replace the conformal framework, but to improve the conditional score-quantile calibration step by replacing the raw conditional quantile curve with a CF-adjusted conditional quantile curve. In this sense, CQACP is best viewed as a sequential conformal calibration method for dependent time series, directly extending the score-based framework used by SPCI (Xu and Xie, 2023).
> > >
> > > We agree that in time-series settings the word conformal should not be interpreted as implying the classical exchangeability-based exact finite-sample guarantee. As in prior sequential conformal methods for dependent time series (Xu and Xie, 2021; Xu and Xie, 2023; Xu, Jiang and Xie, 2024), our result is asymptotic conditional validity under serial dependence, rather than classical exact finite-sample marginal validity. We will make this distinction more explicit in the revision.
> > >
> > > ***Q2: Would it be possible to provide it with ACI-like, weaker, guarantees?***
> > >
> > > ***A2:*** We believe the answer is **yes in principle**, but not for the current fixed-$\alpha$ CQACP exactly as stated. Our current method improves the interval family by stabilizing the conditional score-quantile curve and enforcing nestedness across significance levels, whereas ACI-type methods adapt the deployed level $\alpha_t$ online to target weaker long-run guarantees. A natural hybrid would therefore be to keep the CQACP interval engine unchanged, but replace the fixed $\alpha$ by an online level $\alpha_t$ and deploy $\widehat C_t^{\alpha_t}(X_t)$, where $\alpha_t$ is updated from the realized miscoverage indicator using an ACI-style rule. Such a hybrid would naturally target an ACI-type weaker guarantee (long-run average coverage control), rather than the fixed-$\alpha$ asymptotic conditional-validity theorem proved in the current paper. It is a natural extension of our framework, and we leave a detailed theoretical analysis of its ACI-type guarantees to future work.
> > >
> > > ***Q3: Could the statement that tails benefit disproportionately from the denoising projection be supported numerically by reporting the MSE over $1 - \alpha > 0.1$ or so?***
> > >
> > > ***A3:***  To answer this question, we added a direct oracle-quantile MSE experiment under synthetic designs; see A1 to Reviewer Mgz9 for the detailed setup. To isolate the quantile-calibration effect, we use the oracle conditional mean as the point predictor and compare only the raw SPCI quantile curve against CQACP on the grid $p \in${$0.01,0.02,\ldots,0.99$}.
> > >
> > > We report the reduction of MSE:
> > > $$ \Delta MSE = 1- \frac{MSE_{CQACP}}{MSE_{SPCI}}$$
> > > on the *full grid*, the *center region* ($p\in[0.40,0.60]$), and the *tail region* ($p\in[0.01,0.10]\cup[0.90,0.99]$).
> > >
> > > ***The resulting relative MSE reductions are $14.99$% on the full grid, $2.71$% in the center region, and $19.34$% in the tail region under the heavy-tail design, and $14.31$%, $3.01$%, and $18.26$%, respectively, under the changing-tail design.*** Thus, the improvement is clearly largest in the tail region, directly supporting our statement that the denoising effect of CQACP is strongest where the base conditional quantile estimates are most unstable.
> > >
> > > | Region | Method | Heavy-tail MSE | s.d. | $\Delta$MSE | s.d. | Changing-tail MSE | s.d. | $\Delta$MSE | s.d. |
> > > |--------|--------|----------------|------|-------------|------|-------------------|------|-------------|------|
> > > | Full grid | SPCI  | 0.086 | 0.061 | - | - | 0.089 | 0.095 | - | - |
> > > | Full grid | CQACP | 0.073 | 0.053 | 14.99% | 2.36% | 0.075 | 0.077 | 14.31% | 2.01% |
> > > | Center region| SPCI  | 0.016 | 0.007 | - | - | 0.021 | 0.012 | - | - |
> > > | Center region| CQACP | 0.015 | 0.006 | 2.71% | 1.88% | 0.019 | 0.009 | 3.01% | 1.06% |
> > > | Tail region| SPCI  | 0.282 | 0.214 | - | - | 0.253 | 0.235 | - | - |
> > > | Tail region| CQACP | 0.227 | 0.184 | 19.34% | 3.93% | 0.204 | 0.205 | 18.26% | 3.45% |
> > >
> > > ***We hope that our responses have adequately addressed your further concerns, and we would sincerely appreciate your reconsideration of the evaluation.***
> > >
> > > ***Reference***
> > >
> > > Xu C, Xie Y. Conformal prediction interval for dynamic time-series. ICML, 2021
> > >
> > > Xu C, Xie Y. Sequential predictive conformal inference for time series. ICML, 2023
> > >
> > > Xu C, Jiang H, Xie Y. Conformal prediction for multi-dimensional time series by ellipsoidal sets. ICML, 2024

---

### Official Review · Reviewer_Mgz9 · 2026-03-08

**Soundness:** 4
**Presentation:** 3
**Significance:** 3
**Originality:** 4
**Overall Recommendation:** 5
**Confidence:** 3

**Summary:**

This paper studies sequential conformal prediction for time series and focuses on two practical shortcomings of existing methods: 1) unstable tail estimation and 2) non-nested prediction intervals across levels. The proposed CQACP addresses these issues by treating the conditional quantile function of the nonconformity score as a structured object. It fits a Cornish–Fisher approximation over a grid of quantile levels, enforces monotonicity of the adjusted quantile curve, and uses a shared asymmetry parameter across significance levels to produce nested, non-crossing prediction intervals. The experiment results show better coverage rate and efficiency.

**Compliance With Llm Reviewing Policy:**

Affirmed.

**Key Questions For Authors:**

Please see weakness.

**Limitations:**

yes

**Strengths And Weaknesses:**

Strength:

- The motivation is clear and meaningful, targeted at pain points of existing methods.
- The proposed solution is clean and natural, which shares information across the quantile grid by fitting a structured CF approximation. The idea of utilizing conditional moments is universal and robust under most settings.
- Writing of this paper is well-organized and experiment results are strong.

Weakness:

- If I had to identify one main limitation, it would arise from the CF approximation component. Relying on this approximation probably leads to misspecification under some extreme tail distribution, e.g., heavy-tailed or changing tail behavior. A complementary synthetic experiment would suffice to justify the method.
- As discussed, the paper is designed for single-dimensional and single-step time series, extending to complicated settings would bring challenge to the current framework, especially the nesting property.

---

> ### Author Rebuttal · Authors · 2026-03-29
>
> We thank you very much for the positive assessment and the constructive suggestions. Our point-by-point responses are below.
>
> ***Q1: If I had to identify one main limitation, it would arise from the CF approximation component. Relying on this approximation probably leads to misspecification under some extreme tail distribution, e.g., heavy-tailed or changing tail behavior. A complementary synthetic experiment would suffice to justify the method.***
>
> ***A1:*** We agree that a low-order CF approximation may be misspecified under sufficiently heavy or dynamic tails. To directly address this concern, we add a complementary synthetic experiment with two challenging settings: (1) heavy-tailed Student-t innovations with GARCH dynamics, and (2) changing-tail behavior induced by dynamic Student-t degrees of freedom. Specifically, we generate $Yₜ = μₜ + εₜ$ with an AR-type conditional mean and GARCH(1,1) volatility $σ_t^2 = 0.05 + 0.1ε_{t-1}^2 + 0.85σ_{t-1}^2$. In the heavy-tail design, $zₜ$ ~ rescaled $t_5$. In the changing-tail design, the degrees of freedom alternate between 20 and 5 every 250 time points. For each design, we generate 2500 observations, use a chronological 80%/20% train/test split, and repeat over 100 Monte Carlo runs. We keep the same implementation as in Section 6 to generate the two tables below:
>
> ### Table 1: $\alpha$=10%
>
> | Method | Heavy-tail Coverage | (std) | Heavy-tail Width | (std) | Changing-tail Coverage | (std) | Changing-tail Width | (std) |
> |--------|----------------------|------|------------------|------|------------------------|------|---------------------|------|
> | EnbPI  | 85.0%                | 3.4% | 2.72             | 0.25 | 84.9%                  | 2.7% | 2.82                | 0.25 |
> | SPCI   | 88.7%                | 2.4% | 2.71             | 0.30 | 87.9%                  | 2.2% | 2.78                | 0.30 |
> | CQACP  | 89.7%                | 2.3% | 2.76             | 0.30 | 89.6%                  | 2.1% | 2.84                | 0.30 |
>
> ### Table 2: $\alpha$=5%
> | Method | Heavy-tail Coverage | (std) | Heavy-tail Width | (std) | Changing-tail Coverage | (std) | Changing-tail Width | (std) |
> |--------|----------------------|------|------------------|------|------------------------|------|---------------------|------|
> | EnbPI  | 90.6%                | 2.5% | 3.50             | 0.34 | 90.2%                  | 2.3% | 3.53                | 0.38 |
> | SPCI   | 92.4%                | 1.8% | 3.36             | 0.39 | 92.8%                  | 1.8% | 3.39                | 0.41 |
> | CQACP  | 94.6%                | 1.6% | 3.39             | 1.5% | 94.7%                  | 1.5% | 3.39                | 0.42 |
>
> Across both designs, CQACP achieves coverage closest to $1-\alpha$ at both $α$ = 10% and $α$ = 5%. The absolute coverage error of CQACP is only 0.3%–0.4%, compared with 1.3%–2.6% for SPCI and 4.4%–5.1% for EnbPI. These results suggest that the CF adjustment in CQACP remains effective for the tail quantiles relevant to conformal prediction even when the nonconformity score distribution is heavy-tailed or exhibits changing tail behavior over time.
>
> ***Q2: As discussed, the paper is designed for single-dimensional and single-step time series, extending to complicated settings would bring challenge to the current framework, especially the nesting property.***
>
> ***A2:*** We agree that the current paper studies univariate, one-step-ahead prediction intervals. Meanwhile, CQACP is modular and can be extended to more complicated settings.
>
> For multi-step forecasting, Xu and Xie (2023) propose a divide-and-conquer extension of SPCI that trains different horizon-specific predictors and quantile models. A natural extension is to replace each horizon-specific raw quantile model by a CQACP-adjusted curve. Because the adjusted quantile curve is monotone and the asymmetry ratio is shared across confidence levels, the resulting intervals remain nested in α for each fixed horizon. The multi-step SPCI construction provides the horizon-wise sequential framework, while CQACP provides the stabilized and nested quantile calibration.
>
> For multivariate response, Xu, Jiang and Xie (2024) map the vector residual $\hat{ε}_t$  to a scalar ellipsoidal score and then sequentially calibrate its quantiles to form ellipsoidal prediction regions. This reduction is directly compatible with CQACP, because our adjustment acts on a scalar quantile curve. One can replace the scalar score-quantile estimator in MultiDimSPCI by a CQACP-adjusted score curve. Since nesting is enforced at the scalar-score level, it is inherited by the resulting ellipsoidal regions.
>
> Due to space limitations, we leave these directions for future work.
>
> **We hope our responses and planned revisions have successfully addressed your concerns.**
>
> ### Reference
> Xu C, Xie Y. Sequential predictive conformal inference for time series. ICML, 2023
>
> Xu C, Jiang H, Xie Y. Conformal prediction for multi-dimensional time series by ellipsoidal sets. ICML, 2024

---

> > ### Author Rebuttal · Reviewer_Mgz9 · 2026-04-05
> >
> > Thank you for addressing my concerns. I have no further questions.

---

> > > ### Author Response · Authors · 2026-04-07
> > >
> > > We sincerely appreciate your thoughtful evaluation and encouraging feedback. We are delighted to hear that our rebuttal has satisfactorily addressed your questions, and we are especially grateful for your recommendation to accept the paper.

---

### Official Review · Reviewer_hSWN · 2026-03-11

**Soundness:** 3
**Presentation:** 2
**Significance:** 3
**Originality:** 3
**Overall Recommendation:** 5
**Confidence:** 3

**Summary:**

This work presents a novel conformal prediction method for time series. The method is distinguished by its ability to guarantee smooth, nested prediction intervals for varying confidence intervals. The method adjusts conditional quantile predictions by fitting a Cornish-Fisher expansion to the predictions. The Cornish-Fisher expansion replaces the conditional quantile predictions, which guarantees "smoothness" of the prediction intervals. To guarantee nested prediction intervals, the CF expansion fitting problem is modified with a set of linear constraints. The authors prove theoretical results about their approach (proving asymptotic conditional coverage and improvement in MSE) and demonstrate its performance in several experiments comparing it to strong benchmark methods.

**Compliance With Llm Reviewing Policy:**

Affirmed.

**Final Justification:**

The authors have committed to fixing several major issues in the presentation of their work in the final version. I trust that they will do so, and raise my score accordingly.

**Key Questions For Authors:**

1. Is the benefit of smoothness and nested intervals intrinsic or extrinsic? I.e., does it benefit prediction accuracy itself, or does it benefit downstream uses of the prediction intervals?
2. What are specific examples of when non-nested or non-smooth prediction intervals are problematic or unacceptable?
3. What is a precise definition of exchangeability?
4. What is the significance of $\hat{f}$, the pre-trained prediction model, in conformal prediction?

**Limitations:**

Yes

**Strengths And Weaknesses:**

My assessment is that this work is technically sound - I think the core idea of using Cornish-Fisher expansion to regularize conditional quantile predictions at different confidence levels is quite creative and interesting. Furthermore, the ability to guarantee nested prediction intervals is quite useful, and the performance of the model seems good relative to benchmark models that apparently do not have the same smoothness or nestedness guarantees.

My biggest concern is with the quality of the presentation. Aside from small grammatical errors and inconsistencies, the paper feels slightly disorganized and redundant in places, making it harder to read. In particular, the introduction and literature review (sections 1 and 2) suffer from a lack of mathematical descriptions of the main idea and of related works, and overly rely on lengthy natural language descriptions of mathematical ideas. I found little value in reading these sections, especially given that many of the same ideas are rehashed with mathematical descriptions in sections 3 and 4 (making sections 1 and 2 somewhat redundant). Furthermore, sections 1 and 2 use lots of technical jargon without defining it until later on (e.g. exchangeability, non-conformity score, etc.). The authors also occasionally make statements without giving a citation or proof - e.g. stating "Needless to say, time series data violate the exchangeability assumption due to serial dependence and distribution shift." In sections 3 and 4 and beyond, the quality and clarity of the writing seems to improve, and the mathematical exposition of key ideas greatly helps the reader - however, it is still confusing that related works such as EnbPI and SPCI are being reviewed in section 4, which is titled "Methodology." In my opinion, these descriptions belong in the related work section. Similarly, the description of Cornish-Fisher expansion perhaps deserves its own "background" subsection. Lastly, when presenting the experimental results, it would be good to highlight the best-performing method in each row / experiment (especially if one method dominates the others or is dominated by CQACP).

Aside from these issues with the presentation, I believe the work is of high quality and the idea is very interesting, as I have said. I would consider raising my score if some of the presentation issues were addressed.

---

> ### Author Rebuttal · Authors · 2026-03-29
>
> We thank you very much for the constructive comments. Our point-by-point responses are below.
>
> ***Q1: My biggest concern is with the quality of the presentation.***
>
> ***A1:*** Thank you very much for the detailed feedback. We totally agree that the presentation can be improved, especially in terms of organization and reducing redundancy. In the revision, we will streamline Sections 1–2, add a concise mathematical preview of the main idea earlier in the paper, and define technical terms (e.g., exchangeability) when they first appear. We will also move the descriptions of EnbPI/SPCI out of the "Methodology" section into related work, add a short background subsection on the Cornish–Fisher expansion, and make the methodology section focus more cleanly on CQACP itself. In addition, we will correct grammatical inconsistencies, add missing qualifications where needed, and improve the result tables by more clearly highlighting the strongest-performing methods. We appreciate this suggestion very much and agree that these planned changes would substantially improve readability.
>
> ***Q2: Is the benefit of smoothness and nested intervals intrinsic or extrinsic? I.e., does it benefit prediction accuracy itself, or does it benefit downstream uses of the prediction intervals?***
>
> ***A2:*** The benefits are both intrinsic and extrinsic.
>
> For intrinsic benefits,  the smoothness of the adjusted conditional quantile curve results from the information-sharing mechanism formalized in Theorem 1: Projecting the base quantile estimators onto a low-dimensional CF space reduces estimation noise at each quantile level, especially in the tails. This improved quantile estimation accuracy translates directly into better-calibrated prediction intervals. At $α$ = 10%, CQACP attains coverages around 90.0% across the four datasets, whereas SPCI ranges from 87.9% to 89.3% (Tables 1–2). This improvement is consistent with an intrinsic benefit of the denoising mechanism, not merely a downstream advantage.
>
> For extrinsic benefits, the nestedness property ensures that prediction intervals are coherent across significance levels, which is critical for downstream decision-making. When a practitioner evaluates uncertainty at multiple confidence levels simultaneously, non-nested intervals are logically contradictory and operationally unusable. The smoothness of intervals also improves interpretability: Erratically fluctuating interval boundaries (as seen in SPCI in Figure 1) complicate visual inspection and automated monitoring systems.
>
> ***Q3: What are specific examples of when non-nested or non-smooth prediction intervals are problematic or unacceptable?***
>
> ***A3:*** For financial regulation, under the Basel accords, banks must report Value-at-Risk at multiple confidence levels (e.g., 97.5% and 99%) for market risk capital calculations. A prediction interval system that produces a narrower 99% interval than the 97.5% interval is logically contradictory and would raise immediate flags in regulatory review. In the revision, we will use this example to illustrate the issue of non-nested or non-smooth prediction intervals in Section 1.
>
> ***Q4: What is a precise definition of exchangeability?***
>
> ***A4:*** A finite sequence $\{Z_1, …, Z_n\}$ is exchangeable if its joint distribution is invariant under any permutation: for every permutation $\pi$ of {$1, …, n$}, $(Z_1, …, Z_n) \overset{d}{=} (Z_{π(1)}, …, Z_{π(n)})$. In classical conformal prediction, this is typically assumed for the data points $(X_i, Y_i)$ or equivalently for the conformity scores. We acknowledge that this definition was not provided in the current draft and will add it in the revision.
>
> ***Q5: What is the significance of $\hat{f}$, the pre-trained prediction model, in conformal prediction?***
>
> ***A5:*** The pre-trained prediction model $\hat{f}$ serves two roles. First, it provides the point forecast $\hat{Y}_t = \hat{f}(X_t)$ that centers the prediction interval. Second, it defines the nonconformity score $\hat{\epsilon}_t = Y_t − \hat{f}(X_t)$, which measures how surprising each observation is relative to the prediction. The conformal machinery then calibrates the interval around $\hat{f}(X_t)$ by estimating quantiles of the nonconformity score distribution. A key property of conformal prediction is that $\hat{f}$ can be treated as a black-box predictor: The coverage guarantee holds regardless of how $\hat{f}$ is constructed, though a better $\hat{f}$ typically leads to narrower intervals. In the time-series setting, a good $\hat{f}$ also makes conditional quantile estimation easier and more accurate.
>
> In CQACP, $\hat{f}$ is fixed and pre-trained on in-sample data. CQACP operates entirely on the residuals, modeling their conditional quantile function via the CF adjustment. This design means CQACP can be paired with any user-specified prediction model.
>
> **We hope our responses and planned revisions have successfully addressed your concerns.**

---

> > ### Author Rebuttal · Reviewer_hSWN · 2026-04-02
> >
> > The authors addressed my primary concerns, and I hope they will be able to suitably alter the final presentation/structure of the paper.

---

> > > ### Author Response · Authors · 2026-04-03
> > >
> > > Thank you for your helpful comments again. We are glad to hear that we have addressed your primary concerns, while the remaining concerns are mainly related to the presentation/structure of the paper. As we noted in our earlier response, we fully agree that the presentation and structure of this paper can be improved. In the revision, we plan to streamline Sections 1–2, provide a concise mathematical preview earlier in the paper, define technical terms when they first appear, reorganize background material to reduce redundancy, and improve the clarity of the result tables and overall exposition.
> > >
> > > As ICML does not allow us to revise the manuscript during the rebuttal phase, we are unable to incorporate these changes directly at this stage. Nevertheless, we take this issue seriously and will carefully implement these improvements in the revised version. Given that your remaining concern is about presentation/structure rather than the core technical content, we sincerely hope you may reconsider your evaluation.

---

### Official Review · Reviewer_A3Pb · 2026-03-12

**Soundness:** 3
**Presentation:** 3
**Significance:** 3
**Originality:** 3
**Overall Recommendation:** 3
**Confidence:** 3

**Summary:**

This paper addresses two common issues in conformal prediction for time series. First, tail quantile estimation is often noisy, which causes the interval width to fluctuate sharply over time. Second, estimating different significance levels separately can easily lead to quantile crossing, which in turn makes the resulting prediction intervals non-nested. To address these issues, the paper proposes CQACP. The method evaluates a base conditional quantile learner over a grid of quantile levels, then applies a Cornish–Fisher approximation to structurally adjust the entire conditional quantile curve, together with a monotonicity constraint, so as to produce smoother, more stable, and nested prediction intervals across confidence levels.

**Compliance With Llm Reviewing Policy:**

Affirmed.

**Key Questions For Authors:**

1. How does the choice of K in the Cornish–Fisher adjustment affect performance?
2. In Line 6 of the algorithm, how is the index j determined?
3. The notation for subscripts/superscripts appears inconsistent and should be standardized.
4. Is the method essentially only a smooth approximation to the quantile function?
5. I would recommend that the authors add a more direct discussion in the main text of why Cornish–Fisher adjustment is more appropriate than simply applying monotone rearrangement. It would also strengthen the argument if the paper more clearly separated the contribution of information sharing across quantiles, which improves stability, from the contribution of the monotonicity constraint, which guarantees nestedness.

**Limitations:**

Yes. The authors have adequately discussed the limitations and potential negative societal impacts of their work.

**Strengths And Weaknesses:**

The main claims of the paper are as follows: by adjusting the conditional quantile curve of the nonconformity scores as a whole, rather than estimating each quantile independently, the method can reduce fluctuations in tail quantiles; by imposing a monotonicity constraint, it can avoid quantile crossing and naturally yield nested intervals; and under suitable regularity conditions, the method still preserves asymptotic conditional validity for time series.

The main methodological contribution does not lie in redefining the conformal framework itself, but rather in performing a structured post-processing step on the conditional quantile curve. Compared with methods such as SPCI, the modification introduced by CQACP is quite focused: it retains the main framework of sequential conformal prediction, while incorporating a Cornish–Fisher-based adjustment at the key quantile estimation stage.

The theoretical component is an important strength of the paper. The authors not only claim asymptotic conditional coverage, but also discuss how quantile adjustment improves the accuracy of conditional quantile estimation. Compared with many works that are primarily engineering refinements, this paper attempts to explain why cross-quantile smoothing does not undermine, and may in fact support, the validity of conformal prediction in time-series settings. This is a meaningful contribution.

From the description in the paper, the experiments emphasize three findings: (1) coverage remains accurate; (2) the interval width is often smaller; and (3) the intervals are smoother and nested across multiple significance levels. However, the experimental section could be strengthened further. In particular, I would recommend adding ablations for the Cornish–Fisher adjustment module itself, such as: removing the monotonicity constraint, applying smoothing without the Cornish–Fisher adjustment, or training only at a single α level. It would also be useful to examine whether the gain of CQACP remains consistent when the underlying quantile learner is either particularly strong or particularly weak.

The paper identifies a genuine and practically important problem in sequential conformal prediction; the method is compact and not excessively complicated; and the theoretical and experimental narratives are well aligned.

The practical benefit of the method may depend considerably on the quality of the underlying quantile learner.

---

> ### Author Rebuttal · Authors · 2026-03-29
>
> We thank you very much for the detailed and constructive comments. Our point-by-point responses are below.
>
> ***Q1: How does the choice of $K$ in the Cornish–Fisher adjustment affect performance?***
>
> ***A1***: The truncation order $K$ controls a bias–variance trade-off. A larger $K$ reduces CF truncation bias by incorporating higher-order conditional moments, but increases the estimation variance. Theorem 1 quantifies this: the MSE gain is at least $\sigma^2(1 - K/n) - C K^{-2s}$. In practice, $K$ is selected via cross-validation, defined in (Eq. 15), over {$3,..,10$}. In this rebuttal, we provide a sensitivity analysis on the Realized Volatility dataset and report the related results below:
> | K | Coverage | (std) | Width | (std) |
> |---|----------|-------|-------|-------|
> | 3 | 88.7%    | 0.3%  | 0.8%  | 0.0%  |
> | 4 | 90.1%    | 0.2%  | 0.9%  | 0.0%  |
> | 5 | 90.1%    | 0.4%  | 0.9%  | 0.0%  |
> | 6 | 90.4%    | 0.6%  | 0.9%  | 0.0%  |
>
> The table shows that coverage and width are stable across $K \in $ {$3, \dots, 6$}, and the proposed cross-validation reliably selects the value $K = 4$ on this dataset.
>
> ***Q2: In Line 6 of the algorithm, how is the index $j$ determined?***
>
> ***A2:*** In Line 6 of Algorithm 1, $j$ is a deterministic time index ranging from $w+1$ to $T$, iterating over all in-sample time points where a full residual window of length $w$ is available. The collection $D = {(E_j^w, \hat{\epsilon}_{j}) : j = w+1, …, T}$ forms the training dataset for the base conditional quantile learner. We will state this more explicitly in the revision.
>
> ***Q3: The notation for subscripts/superscripts appears inconsistent and should be standardized.***
>
> ***A3:*** We agree. In the revision, all indices and parameters $(t, K, p, α)$ will appear as subscripts, with only "adj" as a superscript. For example, $Ĉ_{t-1}^{α}(X_t)$ will be replaced by $Ĉ_{α,t-1}(X_t)$.
>
> ***Q4: Is the method essentially only a smooth approximation to the quantile function?***
>
> ***A4:*** CQACP is substantively more than generic smoothing for three reasons. (1) The CF approximation performs structured denoising by projecting the base quantile curve onto a low-dimensional moment-parameterized space, pooling information across quantile levels to stabilize tail estimates. Theorem 1 shows that the MSE gain is governed by variance removal through projection, not merely smoothness, and holds even with biased base conditional quantile estimators (Assumption 3, Lemma 1). (2) The CF coefficients $\theta_{t,K}$ encode time-varying conditional moments (e.g. location, scale, skewness, kurtosis), providing interpretable distributional summaries that kernel or spline smoothing does not. (3) A shared asymmetry parameter (Eq. 12–13) ensures nested prediction intervals across significance levels, which is a structural design property, not a smoothing consequence.
>
> ***Q5: I would recommend that the authors add a more direct discussion in the main text of why Cornish–Fisher adjustment is more appropriate than simply applying monotone rearrangement. It would also strengthen the argument if the paper more clearly separated the contribution of information sharing across quantiles, which improves stability, from the contribution of the monotonicity constraint, which guarantees nestedness.***
>
> ***A5:***
> We agree and will add a dedicated discussion. Monotone rearrangement (Chernozhukov et al., 2010) is a principled post-processing method for enforcing non-crossing quantile curves. Prior work shows that, when the target curve is monotone, rearrangement can improve the estimate in common metrics. Our point here is different: Rearrangement is an ex post monotonicity correction, whereas the CF projection additionally pools information across the full quantile grid through a structured moment-parameterized family. In our framework, this projection step is what yields the MSE improvement in Theorem 1, while the monotonicity constraint together with the shared asymmetry ratio $\hat{\rho}_t$ is what guarantees non-crossing and nestedness. We verify this empirically on the Realized Volatility dataset in the following table:
>
> | Method  | Coverage | (std) | Width | (std) |
> |---------|----------|-------|-------|-------|
> | SPCI    | 87.4%    | 1.3%  | 0.8%  | 0.0%  |
> | SPCI-MR | 88.2%    | 1.1%  | 0.8%  | 0.0%  |
> | CQACP   | 89.9%    | 0.4%  | 0.8%  | 0.0%  |
>
> From this table, SPCI-MR (SPCI + Monotone Rearrangement) improves coverage only modestly over SPCI, whereas CQACP achieves coverage closer to the nominal level with lower variability, suggesting that cross-quantile information sharing, rather than monotonicity correction alone, is the main source of gain. We will make this separation explicit in Section 4 of the revision.
>
> **We hope our responses and planned revisions have successfully addressed your concerns.**
>
> ### Reference
> CHERNOZHUKOV, V., et al. “Quantile and probability curves without crossing.” Econometrica, 2010, 78(3): 1093-1125.

---

> > ### Author Rebuttal · Reviewer_A3Pb · 2026-04-05
> >
> > I want to know how you solve the constrained problem of equation 12? Will this limit the expressiveness of the model?

---

> > > ### Author Response · Authors · 2026-04-05
> > >
> > > Thank you for this helpful question. Equation (12) is a standard *convex constrained least-squares* problem. In matrix form, it can be written as
> > >
> > > $$
> > > \min_{\theta\in\mathbb{R}^K}\||\widehat q_t - Z_K\theta\||_2^2
> > > \quad\text{s.t.}\quad
> > > A\theta \ge 0,
> > > $$
> > >
> > > where the $i$-th row of $Z_K$ is $\psi_K(z_{p_i})^\top$, and each row of $A$ is $\psi_K'(z_j))^\top$ for some $z_j \in \mathcal Z$. Thus, the problem is a quadratic program with linear inequality constraints, which can be solved directly by standard off-the-shelf QP solvers (e.g., interior-point methods). Since the dimension $K$ is a fixed constant and is typically very small in our setting, this constrained solve is computationally inexpensive in practice. Operationally, we first compute the unconstrained OLS solution in (9), and only invoke the constrained solve when the OLS fit violates monotonicity.
> > >
> > > We do not view the monotonicity constraint as materially reducing expressiveness relative to the target object, because a valid quantile curve is monotone by definition. In this sense, the constraint mainly removes finite-sample crossing artifacts caused by estimation noise, rather than excluding meaningful structure of the true quantile function. In fact, the monotonicity constraint mainly enforces coherent non-crossing and nestedness across significance levels, while the main accuracy gain comes from the cross-quantile CF projection, as our simulation results in the first-round reply demonstrate.
> > >
> > > ***We hope that our responses have adequately addressed your further concerns, and we would sincerely appreciate your reconsideration of the evaluation.***

---

### Decision · Program_Chairs · 2026-04-30

**Decision:**

Accept (regular)

**Comment:**

The paper proposes a sequential conformal calibration method for time series that uses Cornish–Fisher expansion to stabilize conditional quantile curve estimation and enforce nested, non-crossing prediction intervals across significance levels. The submission received four reviews with final scores of 5, 5, 4, and 4 after rebuttal revisions, reflecting broad reviewer consensus on acceptance.

Reviewers agreed on several strengths: the motivation targets genuine pain points in sequential conformal prediction, the Cornish–Fisher projection idea is well-motivated, the theoretical contribution (asymptotic conditional validity under serial dependence) is useful, and there are experimental results across multiple real-world datasets.

The main concerns raised during review were all resolved. The terminology concern around "conformal" without exact finite-sample guarantees is consistent with established precedent in the sequential conformal prediction literature (EnbPI, SPCI, MultiDimSPCI), and is not a deficiency specific to this paper. Remaining concerns are presentational and have concrete author-committed revision plans. I recommend acceptance.